# Machine learning-guided realization of full-color high-quantum-yield carbon quantum dots

Huazhang Guo[1,6], Yuhao Lu[2,6], Zhendong Lei[3,6], Hong Bao[1,6], Mingwan Zhang[1], Zeming Wang [1], Cuntai Guan [2] ✉, Bijun Tang [3] ✉, Zheng Liu [3,4,5] ✉ & Liang Wang [1,3] ✉

Carbon quantum dots (CQDs) have versatile applications in luminescence, whereas identifying optimal synthesis conditions has been challenging due to numerous synthesis parameters and multiple desired outcomes, creating an enormous search space. In this study, we present a novel multi-objective optimization strategy utilizing a machine learning (ML) algorithm to intelligently guide the hydrothermal synthesis of CQDs. Our closed-loop approach learns from limited and sparse data, greatly reducing the research cycle and surpassing traditional trial-and-error methods. Moreover, it also reveals the intricate links between synthesis parameters and target properties and unifies the objective function to optimize multiple desired properties like full-color photoluminescence (PL) wavelength and high PL quantum yields (PLQY). With only 63 experiments, we achieve the synthesis of full-color fluorescent CQDs with high PLQY exceeding 60% across all colors. Our study represents a significant advancement in ML-guided CQDs synthesis, setting the stage for developing new materials with multiple desired properties.

Luminescent materials have gained significant attention in recent decades for their versatile optoelectronic applications in fields such as LED, life medicine, and solar cells[1–9]. Among them, carbon quantum dots (CQDs) have emerged as a promising alternative to conventional luminescent materials due to their unique properties, including low cost, environmental friendliness, size tunability, and excellent optical properties[10–15]. However, recent studies have shown that the properties of CQDs are not solely determined by their chemical components but are also heavily influenced by their synthesis conditions[16–20]. For instance, CQDs produced at different reaction temperatures and reaction time may exhibit vastly different luminescent properties. Unfortunately, commonly used synthesis methods for CQDs, such as

the hydrothermal method, often have numerous synthesis parameters like temperature, reaction time, solvent, and catalyst, resulting in an enormous and complex search space. As a result, the preparation of CQDs with desired properties typically requires extensive experimentation in the laboratory. Furthermore, traditional trial-and-error approaches to exploring synthesis methods can easily lead to suboptimal results.

In recent years, the development of machine learning (ML) has provided an exciting new avenue for researchers to overcome the challenges of optimizing the synthesis of CQDs[3,21–28]. As a cutting-edge technology within the field of artificial intelligence, ML has proven to be a valuable tool in identifying complex relationships between

[1]Institute of Nanochemistry and Nanobiology, School of Environmental and Chemical Engineering, Shanghai University, 99 Shangda Road, BaoShan District, Shanghai 200444, China. [2]College of Computing and Data Science, Nanyang Technological University, 50 Nanyang Avenue, Singapore 639798, Singapore. [3]School of Materials Science and Engineering, Nanyang Technological University, 50 Nanyang Avenue, Singapore 639798, Singapore. [4]CINTRA CNRS/NTU/THALES, UMI 3288, Research Techno Plaza, 50 Nanyang Drive, Border X Block, Level 6, Singapore 637553, Singapore. [5]Institute for Functional Intelligent Materials, National University of Singapore, Singapore, Singapore. [6]These authors contributed equally: Huazhang Guo, Yuhao Lu, Zhendong Lei, Hong Bao. ✉e-mail: ctguan@ntu.edu.sg; bjtang@ntu.edu.sg; z.Liu@ntu.edu.sg; wangl@shu.edu.cn

material descriptors and desired properties, especially in high-dimensional and complex search spaces[29–33]. In our previous work, we have successfully introduced ML into CQD synthesis, not only to predict the optimal reaction conditions to obtain CQDs with superior optical properties, but also to establish the correlation between the properties of CQDs and their synthesis conditions[21]. Subsequently, other research groups have adopted various ML algorithms to advance the development of CQDs[21–28]. However, current research mainly focuses on a single target property of materials, whereas real-world material design often demands multiple property criteria to be met. For instance, the optical characteristics of fluorescent CQDs are jointly determined by photoluminescence (PL) wavelength and quantum yields (PLQY). Even when focusing on just one property, previous studies still required substantial amount of data ($\geq 500$) from experiments or calculations to achieve their objectives[21,29,34,35]. The vast search space coupled with multiple target properties makes finding optimal synthesis conditions exponentially more challenging. Therefore, it is imperative to develop an efficient optimization strategy that can simultaneously address multiple desired properties while leveraging the power of ML models.

In this study, we present a novel multi-objective optimization (MOO) strategy to intelligently guide the hydrothermal synthesis of CQDs, integrating state-of-the-art ML techniques with iterative experiments and characterization. Our closed-loop approach can learn from limited and sparse data, enabling us to uncover the hidden relationships between synthesis parameters and various target properties. Moreover, our approach unifies the objective function for multiple desired optical properties, including full-color PL wavelength and high PLQY. Full-color fluorescent CQDs with high PLQY (>60% for all colors) have been achieved within merely 20 iterations, showcasing the superior efficiency and effectiveness of our ML-based approach.

## Results

### Workflow of ML-guided synthesis of CQDs

Synthesis parameters have great impacts on the target properties of resulting samples. However, it is intricate to tune various parameters for optimizing multiple desired properties simultaneously. Our ML-integrated MOO strategy tackles this challenge by learning the complex correlations between hydrothermal/solvothermal synthesis parameters and two target properties of CQDs in a unified MOO formulation, thus recommending optimal conditions that enhance both

properties simultaneously. The overall workflow for the ML-guided synthesis of CQDs is shown in Fig. 1 and Supplementary Fig. 1. The workflow primarily consists of four key components: database construction, multi-objective optimization formulation, MOO recommendation, and experimental verification.

Using a representative and comprehensive synthesis descriptor set is of vital importance in achieving the optimization of synthesis conditions[36]. We carefully selected eight descriptors to comprehensively represent the hydrothermal system, one of the most common methods to prepare CQDs. The descriptor list includes reaction temperature (T), reaction time (t), type of catalyst (C), volume/mass of catalyst ($V_C$), type of solution (S), volume of solution ($V_S$), ramp rate ($R_r$), and mass of precursor ($M_p$). To minimize human intervention, the bounds of synthesis parameters are determined primarily by the constraints of the synthesis methods and equipment used, instead of expert intuition. For instance, in employing hydrothermal/solvothermal method to prepare CQDs, as the reactor inner pot is made of polytetrafluoroethylene material, the usage temperature should be ≤ 220 °C. Moreover, the capacity of the reactor inner pot used in the experiment is 25 mL, with general guidance of not exceeding 2/3 of this volume for reactions. Therefore, in this study, the main considerations of experimental design are to ensure experimental safety and accommodate the limitations of equipment. These practical considerations naturally led to a vast parameter space, estimated at 20 million possible combinations, as detailed in Supplementary Table 1. Briefly, the 2,7-naphthalenediol molecule along with catalysts such as $H_2SO_4$, HAc, ethylenediamine (EDA) and urea, were adopted in constructing the carbon skeleton of CQDs during the hydrothermal or solvothermal reaction process (Supplementary Fig. 2). Different reagents (including deionized water, ethanol, N,N-dimethylformamide (DMF), toluene, and formamide) were used to introduce different functional groups into the architectures of CQDs, combined with other synthesis parameters, resulting in tunable PL emission. To establish the initial training dataset, we collected 23 CQDs synthesized under different randomly selected parameters. Each data sample is labelled with experimentally verified PL wavelength and PLQY (see Methods).

To account for the varying importance of multiple desired properties, an effective strategy is needed to quantitatively evaluate candidate synthesis conditions in a unified manner. A MOO strategy has thus been developed that prioritizes full-color PL wavelength over PLQY enhancement, by assigning an additional reward when maximum

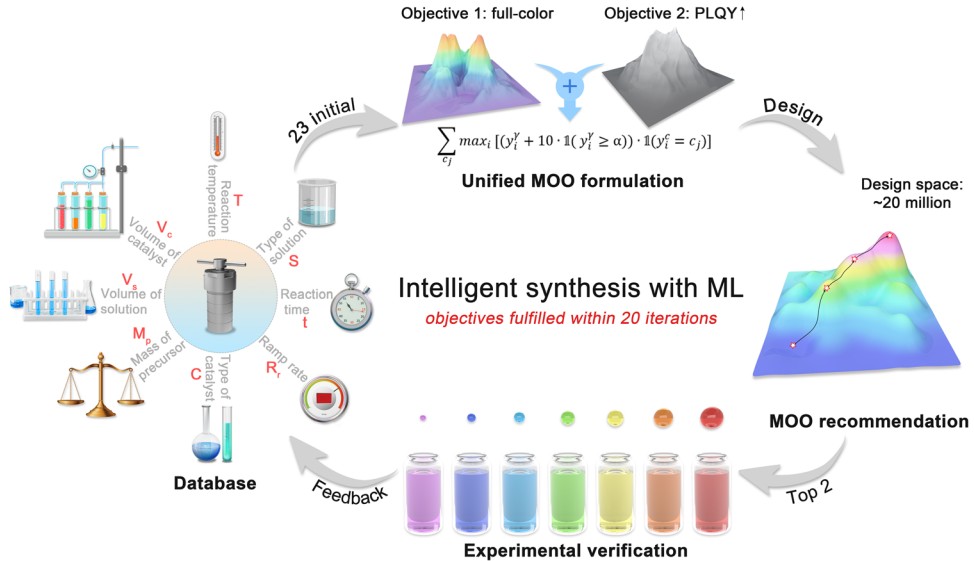

**Fig. 1 | Workflow of ML-guided synthesis of carbon quantum dots (CQDs) with superior optical properties.** It consists of four key components: database construction, multi-objective optimization (MOO) formulation, MOO recommendation, and experimental verification.

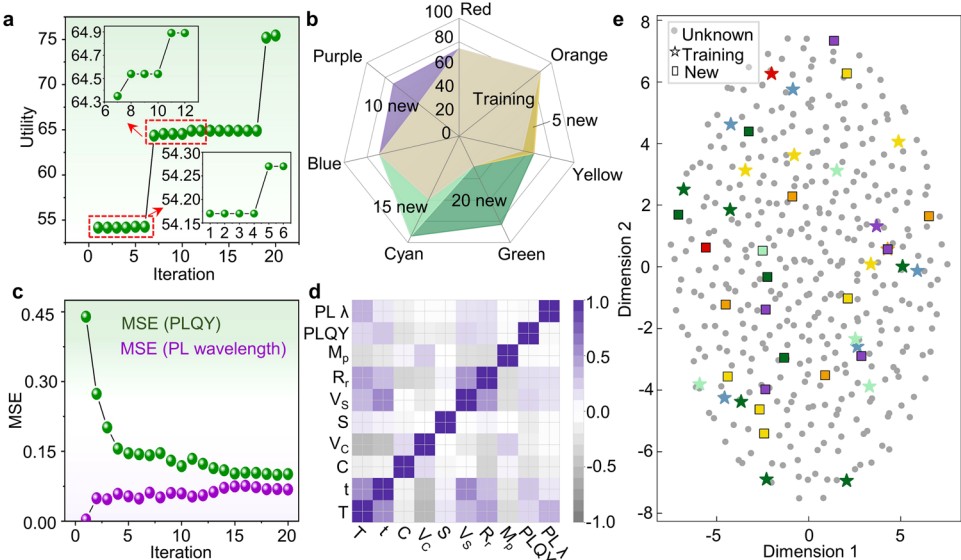

**Fig. 2 | Evaluation of full-color CQDs synthesis guided by ML-based MOO strategy. a** MOO's unified objective utility versus design iterations. **b** Color explored with new synthesized experimental conditions. Value ranges of colors defined by PL wavelength: purple (PL < 420 nm), blue (420 nm ≤ PL < 460 nm), cyan (460 nm ≤ PL < 490 nm), green (490 nm ≤ PL < 520 nm), yellow (520 nm ≤ PL < 550 nm), orange (550 nm ≤ PL < 610 nm), and red (610 nm ≤ PL). It shows that while high PLQY has been achieved for "red", "orange", and "blue" in the initial dataset, the MOO strategy purposefully enhances PLQYs for "yellow", "purple", "cyan", "green" respectively in subsequent synthesized conditions in a group of five. **c** MSE

PLQY of a color surpassing the predefined threshold for the first time. Given $N$ explored experimental conditions, $\{(x_i, y_i^c, y_i^y) | i = (1, 2, \ldots, N)\}$, $x_i$ indicates the $i$-th synthesis condition defined by 8 synthesis parameters, $y_i^c$ and $y_i^y$ indicate the corresponding color label and yield (i.e., PLQY) given $x_i$; $y_i^c \in \{c_1, c_2, \ldots, c_M\}$ for $M$ possible colors, $y_i^y \in [0, 1]$. The unified objective function is formulated as the sum of maximum PLQY for each color label, i.e.,

$$\sum_{c_j} Y_{c_j}^{max}, \tag{1}$$

where $j \in \{1, 2, \ldots, M\}$ and $Y_{c_j}^{max}$ is 0 if $\nexists y_i^c = c_j$; otherwise

$$Y_{c_j}^{max} = \max_i \left[ \left( y_i^y + R \cdot \mathbb{1}\left(y_i^y \ge \alpha\right) \right) \cdot \mathbb{1}\left(y_i^c = c_j\right) \right]. \tag{2}$$

$\mathbb{1}(\cdot)$ is an indicator function that output 1 if true, otherwise outputs 0. The term $R \cdot \mathbb{1}(y_i^y \ge \alpha)$ enforces a higher priority of full-color synthesis, where PLQY for each color shall be at least $\alpha$ ($\alpha = 0.5$ in our case) to have an additional reward of $R$ ($R = 10$ in our settings). $R$ can be any real value larger than 1 (i.e., maximum possible improvement of PLQY for one synthesis condition), to ensure the higher priority of exploring synthesis conditions for colors in which yield has not achieved $\alpha$. We set $R$ to 10, such that the tens digit of unified objective function's value clearly indicates the number of colors with maximum PLQYs exceeding $\alpha$, and the units digit reflects the sum of maximum PLQYs (without the additional reward) for all colors. As defined by the ranges of PL wavelength in Supplementary Table 2, seven primary colors considered in this work are purple (<420 nm), blue (≥420 and <460 nm), cyan (≥460 and <490 nm), green (≥490 and <520 nm), yellow (≥520 and <550 nm), orange (≥550 and <610 nm), and red (≥610 nm), i.e., $M = 7$. Notably, the proposed MOO formulation unifies the two goals of achieving full color and high PLQY into a single objective function, providing a systematical approach to tune synthesis parameters for the desired properties.

between the predicted and real target properties. **d** Covariance matrix for correlation among the 8 synthesis parameters (i.e., reaction temperature T, reaction time t, type of catalyst C, volume/mass of catalyst $V_C$, type of solution S, volume of solution $V_S$, ramp rate $R_r$, and mass of precursor $M_p$) and 2 target properties, i.e., PLQY and PL wavelength (PL λ). **e** Two-dimensional t-distributed stochastic neighbor embedding (t-SNE) plot for the whole search space, including unexplored (circular points), training (star-shaped points), and explored (square points) conditions, where the latter two sets are colored by real PL wavelengths.

The MOO strategy is premised on the prediction results of ML models. Due to the high-dimensional search space and limited experimental data, it is challenging to build models that generalize well on unseen data, especially considering the nonlinear nature of the condition-property relationship[37]. To address this issue, we employed a gradient boosting decision tree-based model (XGBoost), which has proven advantageous in handling related material datasets (see Methods and Supplementary Fig. 3)[30,38]. In addition, its capability to guide hydrothermal synthesis has been proven in our previous work (Supplementary Fig. 4)[21]. Two regression models, optimized with the best hyperparameters through grid search, were fitted on the given dataset, one for PL wavelength and the other for PLQY. These models were then deployed to predict all unexplored candidate synthesis conditions. The search space for candidate conditions is defined by the Cartesian product of all possible values of eight synthesis parameters, resulting in ~20 million possible combinations (see Supplementary Table 1). The candidate synthesis conditions, i.e., unexplored regions of the search space, are further ranked by MOO evaluation strategy with the prediction results.

Finally, the PL wavelength and PLQY values of the CQDs synthesized under the top two recommended synthesis conditions are verified through experiments and characterization, whose results are then augmented to the training dataset for the next iteration of the MOO design loop. The iterative design loops continue until the objectives are fulfilled, i.e., when the achieved PLQY for all seven colors surpasses 50%. In prior studies on CQDs, it's worth noting that only a limited number of CQDs with short-wavelength fluorescence (e.g., blue and green), have reached PLQYs above 50%[39–41]. On the other hand, their long-wavelength counterparts, particularly those with orange and red fluorescence, usually demonstrate PLQYs under 20%[42–44]. Underlining the efficacy of our ML-powered MOO strategy, we have set an ambitious goal for all fluorescent CQDs: the attainment of PLQYs exceeding 50%. The capacity to modulate the PL emission of CQDs holds significant promise for various applications, spanning from bioimaging

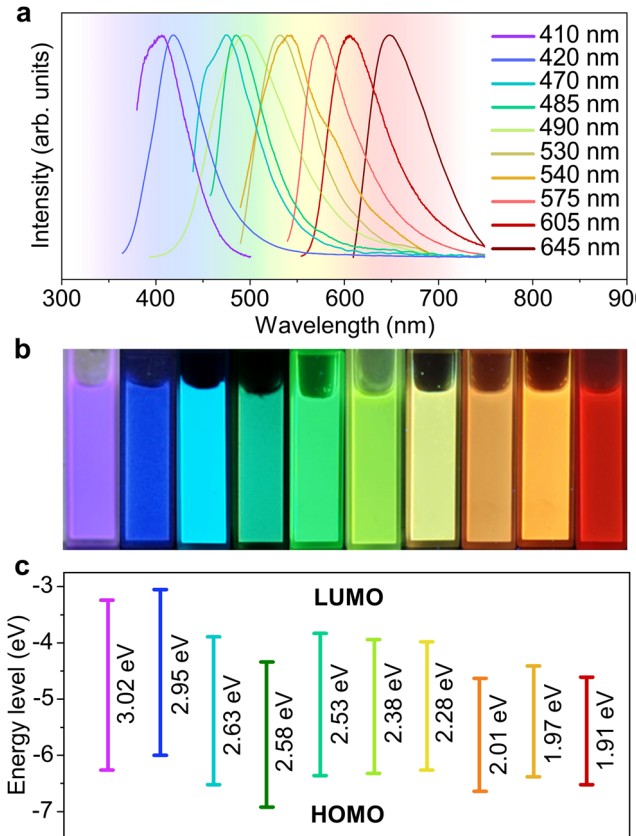

**Fig. 3 | Optical properties of full-color fluorescent CQDs. a** Normalized PL spectra of CQDs. **b** Photographs of CQDs under 365 nm-UV light irradiation. **c** Dependence of the HOMO and LUMO energy levels of CQDs.

and sensing to optoelectronics. Our four-stage workflow is crafted to forge an ML-integrated MOO strategy that can iteratively guide hydrothermal synthesis of CQDs for multiple desired properties, while also constantly improving the models' prediction performance.

## Evaluation of ML-integrated MOO strategy

To assess the effectiveness of our ML-driven MOO strategy in the hydrothermal synthesis of CQDs, we employed several metrics, which were specifically chosen to ascertain whether our proposed approach not only meets its dual objectives but also enhances prediction accuracy throughout the iterative process. The unified objective function described above measures how well the two desired objectives have been realized experimentally, and thus can be a quantitative indicator of the effectiveness of our proposed approach in instructing the CQD synthesis. The evaluation output of the unified objective function after a specific ML-guided synthesis loop is termed as objective utility value. The MOO strategy improves the objective utility value by a large margin of 39.27% to 75.44, denoting that the maximum PLQY in all seven colors exceeds the target of 0.5 (Fig. 2a). Specifically, at iterations 7 and 19, the number of color labels with maximum PLQY exceeding 50% increases by one, resulting in an additional reward of 10 each time. Even on the seemingly plateau, the two insets illustrate that the maximally achieved PLQY is continuously enhanced. For instance, during iterations 8 to 11, the maximum PLQY for cyan emission escalates from 59% to 94%, and the maximum PLQY for purple emission rises from 52% to 71%. Impressively, our MOO approach successfully fulfilled both objectives within only 20 iterations (i.e., 40 guided experiments).

Figure 2b reveals that the MOO strategy systematically explores the synthesis conditions for each color, addressing those that have not

yet achieved the designed PLQY threshold, starting with yellow in the first 5 iterations and ending with green in the last 5 iterations. Notably, within each quintet of 5 iterations, a singular color demonstrates an enhancement in its maximum PLQY. Initially, the PLQY for yellow surges to 65%, which is then followed by a significant rise in purple's maximum PLQY from 44% to 71% during the next set of 5 iterations. This trend continues with cyan and green, where the maximum PLQY escalates to 94% and 83% respectively. Taking into account both the training set (i.e., the first 23 samples) and the augmented dataset, the peak PLQY for all colors exceeds 60%. Several colors approach 70% (including purple, blue, and red), and some are near 100% (including cyan, green, and orange). This further underscores the effectiveness of our proposed ML technique. A more detailed visualization of the PL wavelength and PLQY along each iteration is provided in Supplementary Fig. 5.

The MOO strategy ranks candidate synthesis conditions based on ML prediction; thus, it is vital to evaluate the ML model's performance. Mean squared error (MSE) is employed as the evaluation metric, commonly used for regression, which is computed based on the predicted PL wavelength and PLQY from the ML models versus the experimentally determined values[45]. As shown in Fig. 2c, the MSE of PLQY drastically decreases from 0.45 to approximately 0.15 within just four iterations − a notable error reduction of 64.5%. The MSE eventually stabilizes around 0.1 as the iterative loops progress. Meanwhile, the MSE of PL wavelength remains consistently low, always under 0.1. MSE of PL wavelength is computed after normalizing all values to the range of zero to one for a fair comparison, thus MSE of 0.1 signifies a favorable deviation within 10% between the ML-predicted values and the experimental verifications. This indicates that the accuracies of our ML models for both PL wavelength and PLQY consistently improve, with predictions closely aligning with actual values after enhanced learning from augmented data. This not only demonstrates the efficacy of our MOO strategy in optimizing multiple desired properties but also in refining ML models.

To unveil the correlation between synthesis parameters and target properties, we further calculated the covariance matrix. As illustrated in Fig. 2d, the eight synthesis parameters generally exhibit low correlation among each other, indicating that each parameter contributes unique and complementary information for the optimization of the CQDs synthesis conditions. In terms of the impact of these synthesis parameters on target properties, factors such as reaction time and temperature are found to influence both PL wavelength and PLQY. This underscores the importance for both experimentalists and data-driven methods to adjust them with higher precision. Besides reaction time and temperature, PL wavelength and PLQY are determined by distinct sets of synthesis parameters with varying relations. For instance, the type of solution affects PLQY with a negative correlation, while solution volume has a stronger positive correlation with PLQY. This reiterates that, given the high-dimensional search space, the complex interplay between synthesis parameters and multiple target properties can hardly be unfolded without capable ML-integrated methods.

To visualize how the MOO strategy has navigated in the expansive search space (~20 million) using only 63 data samples, we have compressed the initial training, explored, and unexplored space into two dimensions by projecting them into a new reduced embedding space using t-distributed stochastic neighbor embedding (t-SNE)[46]. As shown in Fig. 2e, discerning distinct clustering patterns by color proves challenging, which emphasizes the intricate task of uncovering the relationship between synthesis conditions and target properties. This complexity further underscores the critical role of a ML-driven approach in deciphering the hidden intricacies within the data. The efficacy of ML models is premised on the quality of training data. Thus, selecting training data that span as large search space as possible is particularly advantageous to models' generalizability[37]. As observed in

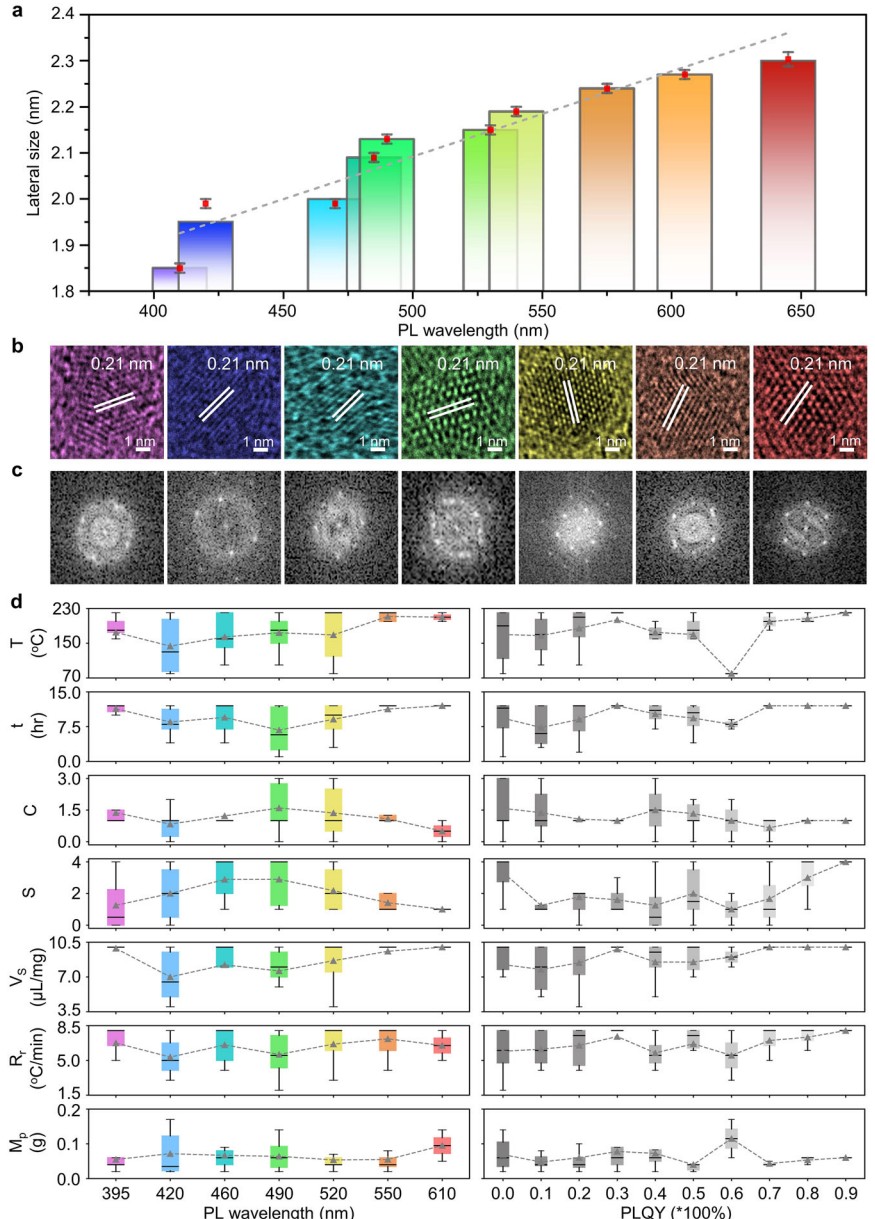

**Fig. 4 | Morphological characterizations and relationship analysis between synthesis parameters and optical properties of full-color fluorescent CQDs.**
**a** The lateral size and color of full-color fluorescent CQDs (inset: dependence of the PL wavelength and the lateral size of full-color fluorescent CQDs). Data correspond to mean ± standard deviation, $n = 3$. **b**, **c** High-resolution TEM images and the fast Fourier transform patterns of p-, b-, c-, g-, y-, o- and r-CQDs, respectively. **d** Boxplots of PL wavelength (left)/PLQY (right) and 7 synthesis parameters of CQDs. $V_C$ is excluded here as its value range is dependent on C, whose relationships with other

parameters are not directly interpretable. The labels at the bottom indicate the minimum value (inclusive) for the respective bins, whereas the bins on the left are the same as the discretization of colors in Supplementary Table 2, the bins on the right are uniform. Each box spans vertically from the 25th percentile to the 75th percentile, with the horizontal line marking the median and the triangle indicating the mean values. The upper and lower whiskers extend from the ends of the box to the minimum and maximum data values.

Fig. 2e, our developed ML models benefit from the randomly and sparsely distributed training data, which in turn encourage the models to further generalize to previously unseen areas in the search space, and effectively guide the searching of optimal synthesis conditions within this intricate multi-objective optimization landscape.

## Optical performance of full-color CQDs synthesized under ML guidance

With the aid of ML-coupled MOO strategy, we have successfully and rapidly identified the optimal conditions giving rise to full-color CQDs with high PLQY. The ML-recommended synthesis conditions that produced the highest PLQY of each color are detailed in the Methods

section. Ten CQDs with the best optical performance were selected for in-depth spectral investigation. The resulting absorption spectra of the CQDs manifest strong excitonic absorption bands, and the normalized PL spectra of the CQDs displayed PL peaks ranging from 410 nm of purple CQDs (p-CQDs) to 645 nm of red CQDs (r-CQDs), as shown in Fig. 3a and Supplementary Fig. 6. This encompasses a diverse array of CQD types, including p-CQDs, blue CQDs (b-CQDs, 420 nm), cyan CQDs (c-CQDs, 470 nm), darkcyan CQDs (dc-CQDs, 485 nm), green CQDs (g-CQDs, 490 nm), yellow-green CQDs (yg-CQDs, 530 nm), yellow CQDs (y-CQDs, 540 nm), orange CQDs (o-CQDs, 575 nm), orange red CQDs (or-CQDs, 605 nm), and r-CQDs. Importantly, PLQY of most of these CQDs were above 60% (Supplementary Table 3), exceeding

the majority of CQDs reported to date (Supplementary Table 4). Corresponding photographs of full-color fluorescence ranging from purple to red light under UV light irradiation are provided in Fig. 3b. Excellent excitation-independent behaviors of the CQDs have been further revealed by the three-dimensional fluorescence spectra (Supplementary Fig. 7). Furthermore, a comprehensive investigation of the time-resolved PL spectra revealed a notable trend. The mono-exponential lifetimes of CQDs progressively decreased from 8.6 ns (p-CQDs) to 2.3 ns (r-CQDs) (Supplementary Fig. 8). This observation signified that the lifetimes of CQDs diminished as their PL wavelength experiences a shift towards the red end of the spectrum[47]. Moreover, the CQDs also demonstrate long-term photostability (>12 hours), rendering them potential candidates for applications in optoelectronic devices that require stable performance over extended periods of time (Supplementary Fig. 9). All the results together demonstrate the high quality and great potential of our synthesized CQDs.

To gain further insights into the properties of the synthesized CQDs, we calculated their bandgap energies using the experimentally obtained absorption band values (Supplementary Fig. 10 and Table 5). It is revealed that the calculated bandgap energies gradually decrease from 3.02 to 1.91 eV from p-CQDs to r-CQDs. In addition, we measured the highest occupied molecular orbital (HOMO) energy levels of the CQDs using ultraviolet photoelectron spectroscopy. As shown in the energy diagram in Fig. 3c, the HOMO values exhibit wave-like variations without any discernible pattern. This result further suggests the robust predictive and optimizing capability of our ML-integrated MOO strategy, which enabled the successful screening of these high-quality CQDs from vast and complex search space using only 40 sets of experiments.

### Structural insights into multicolor CQDs synthesized under ML recommendation

To uncover the underlying mechanism of the tuneable optical effect of the synthesized CQDs, we have carried out a series of characterizations to comprehensively investigate their morphologies and structures (see Methods). X-ray diffraction (XRD) patterns with a single graphite peak at 26.5° indicate a high-degree graphitization in all CQDs (Supplementary Fig. 11)[15]. Raman spectra exhibit a stronger signal intensity for the ordered $G$ band at 1585 cm$^{-1}$ compared to the disordered $D$ band at 1397 cm$^{-1}$, further confirming the high-degree graphitization (Supplementary Fig. 12)[48]. Fourier-transform infrared (FT-IR) spectroscopy was then performed to detect the functional groups in CQDs, which clearly reveals the $-NH_2$ and $N-C$ stretching at 3234 and 1457 cm$^{-1}$, respectively, indicating the presence of abundant $NH_2$ groups on the surface of CQDs, except for orange CQDs (o-CQDs) and yellow CQDs (y-CQDs) (Supplementary Fig. 13)[49]. The $C=C$ aromatic ring stretching at 1510 cm$^{-1}$ confirms the carbon skeleton, while three oxide-related peaks, i.e., $O-H$, $C=O$, and $C-O$ stretching, were observed at 3480, 1580, and 1240 cm$^{-1}$, respectively, due to abundant hydroxyl groups of the precursor. The FT-IR spectrum also shows a stretching vibration band $SO_3$ at 1025 cm$^{-1}$, confirming the additional functionalization of y-CQDs by $SO_3H$ groups.

X-ray photoelectron spectroscopy (XPS) was adopted to further probe the functional groups in CQDs (Supplementary Fig. 14 to 23). XPS survey spectra analysis reveals three main elements in CQDs, i.e., C, O, and N, except o-CQDs and y-CQDs. Specifically, o-CQDs and y-CQDs lack the N element and y-CQDs contains S element. The high-resolution C1s spectrum of CQDs can be deconvoluted into three peaks, including a dominant $C-C/C=C$ graphitic carbon bond (284.8 eV), $C-O/C-N$ (286 eV), and carboxylic $C=O$ (288 eV), revealing the structures of CQDs. The N1s peak at 399.7 eV indicates the presence of $N-C$ bonds, verifying the successful N-doping in the basal plane network structure of CQDs, except o-CQDs and y-CQDs. The separated peaks of O1s at 531.5 and 533 eV indicate the two forms of oxyhydrogen functional groups with $C=O$ and $C-O$, respectively,

consistent with the FT-IR spectra[50]. The S2p band of y-CQDs can be decomposed into two peaks at 163.5 and 167.4 eV, representing $SO_3/2P_{3/2}$ and $SO_3/2P_{1/2}$, respectively[47,51]. Combining the results of structure characterization, the excellent fluorescence properties of the CQDs are attributed to the presence of N-doping, which reduces non-radiative sites of CQDs and promotes the formation of $C=O$ bonds. The $C=O$ bonds play a crucial role in radiation recombination and can increase the PLQY of the CQDs.

To gain deeper insights into the morphology and microstructures of the CQDs, we have then conducted transmission electron microscopy (TEM). The TEM images demonstrate uniformly shaped and monodisperse nanodots, with the gradual increase of average lateral sizes ranging from 1.85 nm for p-CQDs to 2.3 nm for r-CQDs (Fig. 4a and Supplementary Fig. 24), which agrees with the corresponding PL wavelength (Fig. 4a), providing further evidence for the quantum size effect of CQDs (Fig. 4a)[47]. High-resolution TEM images further reveal the highly crystalline structures of CQDs with well-resolved lattice fringes (Fig. 4b-c). The measured crystal plane spacing of 0.21 nm corresponds to the (100) graphite plane, further corroborating the XRD data. Our analysis suggests that the synthesized CQDs possess a graphene-like high-crystallinity characteristic, thereby giving rise to their superior fluorescence performance.

### ML insights into CQDs synthesis

Following the effective utilization of ML in thoroughly exploring the entire search space, we proceeded to conduct a systematic examination of 63 samples using box plots, aiming to elucidate the complex interplay between various synthesis parameters and the resultant optical properties of CQDs. As depicted in Fig. 4d, the synthesis under conditions of high reaction temperature, prolonged reaction time, and low-polarity solvents, tends to result in CQDs with a larger PL wavelength. These findings are consistent with the general observations in the literature, which suggest that the parameters identified above can enhance precursor molecular fusion and nucleation growth, thereby yielding CQDs with increased particle size and high PL wavelength[47,52–55]. Moreover, a comprehensive survey of existing literature implies that precursors and catalysts, typically including electron donation and acceptance, aid in producing long-wavelength CQDs[56,57]. Interestingly, diverging from traditional findings, we successfully synthesized long-wavelength red CQDs under ML guidance, with 2,7-naphthalenediol containing electron-donating groups as the precursor and EDA is known for its electron-donating functionalities as the catalyst. This significant breakthrough questions existing assumptions and offers new insights into the design of long-wavelength CQDs.

Concerning PLQY, we found that catalysts with stronger electron-donating groups (e.g., EDA) led to enhanced PLQY in CQDs, consistent with earlier observations made by our research team[16]. Remarkably, we uncovered the significant impact of synthesis parameters on CQDs' PLQY. In the high PLQY regime, strong positive correlations were discovered between PLQY and reaction temperature, reaction time, and solvent polarity, previously unreported in the literature[58–61]. This insight could be applied to similar systems for PLQY improvement.

Aside from the parameters discussed above, other factors such as ramp rate, the amount of precursor, and solvent volume also influence the properties of CQDs. Overall, the emission color and PLQY of CQDs are governed by complex, non-linear trends resulting from the interaction of numerous factors. It's noteworthy to mention that the traditional methods used to adjust CQDs' properties often result in a decrease in PLQY as the PL wavelength redshifts[4,47,51,54]. However, utilizing AI-assisted synthesis, we have successfully increased the PLQY of the resulting full-color CQDs to over 60%. This significant achievement highlights the unique advantages offered by ML-guided CQDs synthesis and confirms the powerful potential of ML-based methods in effectively navigating the complex relationships among diverse

synthesis parameters and multiple target properties within a high-dimensional search space.

## Discussion

In summary, we have successfully developed and implemented an ML-powered MOO strategy for the hydrothermal synthesis of CQDs. This innovative approach leverages limited and sparse data to uncover the hidden relationships between synthesis parameters and target properties, i.e., PL wavelength and PLQY, while unifying the objective function to optimize for multiple desired properties, i.e., full-color fluorescence and high PLQY. Through only 63 experiments with ML guidance, we have successfully obtained full-color fluorescent CQDs with tunable PL spanning from blue to red light, and remarkable PLQY exceeding 60% for all colors, as evidenced by comprehensive characterizations. These CQDs also exhibit robust long-term fluorescence stability. Additionally, a linear correlation between the particle size of the synthesized CQDs and their corresponding PL wavelength has been identified, illustrating the quantum size effect inherent in CQDs.

Unlike other ML applications in materials synthesis that typically require a large dataset to address a single property, our approach demonstrates remarkable success in optimizing multiple properties with a relatively small number of experiments. This capability not only saves time and resources but also paves the way for early-stage exploration of materials synthesis, opening new possibilities and representing a substantial leap forward compared to traditional trial-and-error methods.

## Methods
### Materials
2,7-naphthalenediol was purchased from Aladdin. Reagent Co., ltd. (China). Urea, ethylenediamine (EDA), $H_2SO_4$, acetic acid, ethanol, N,N-dimethylformamide (DMF), formamide, toluene was purchased from Shanghai Titan Technology Co. Ltd. All chemical reagents were directly used.

### Preparation of full-color fluorescence CQDs
Synthesis of purple CQDs (p-CQDs): 2,7-naphthalenediol (0.04 g) and urea (0.02 g) were dissolved in 10 mL of toluene, the mixture solution was transferred into a poly(tetrafluoroethylene) (Teflon)-lined autoclave in an oven after ultra-sounding for 5 min. Then the reaction was performed with a ramp rate of 8 °C/min at 180 °C for 12 h.

Synthesis of blue CQDs (b-CQDs): 2,7-naphthalenediol (0.04 g) and EDA (170 μL) were added in 10 mL of $H_2O$, the mixture solution was transferred into a poly(tetrafluoroethylene) (Teflon)-lined autoclave in an oven after ultra-sounding for 5 min. Then the reaction was performed with a ramp rate of 7 °C/min at 200 °C for 11 h.

Synthesis of cyan CQDs (c-CQDs): 2,7-naphthalenediol (0.08 g) and urea (0.1 g) were dissolved in 4 mL of $H_2O$, the mixture solution was transferred into a poly(tetrafluoroethylene) (Teflon)-lined autoclave in an oven after ultra-sounding for 5 min. Then the reaction was performed with a ramp rate of 8 °C/min at 160 °C for 4 h.

Synthesis of darkcyan CQDs (dc-CQDs): 2,7-naphthalenediol (0.06 g) and urea (0.06 g) were dissolved in 10 mL of $H_2O$, the mixture solution was transferred into a poly(tetrafluoroethylene) (Teflon)-lined autoclave in an oven after ultra-sounding for 5 min. Then the reaction was performed with a ramp rate of 8 °C/min at 220 °C for 12 h.

Synthesis of green CQDs (g-CQDs): 2,7-naphthalenediol (0.02 g) and urea (0.06 g) were dissolved in 10 mL of ethanol, the mixture solution was transferred into a poly(tetrafluoroethylene) (Teflon)-lined autoclave in an oven after ultra-sounding for 5 min. Then the reaction was performed with a ramp rate of 8 °C/min at 160 °C for 12 h.

Synthesis of yellow green CQDs (yg-CQDs): 2,7-naphthalenediol (0.06 g) and EDA (940 μL) were added in 10 mL of DMF, the mixture solution was transferred into a poly(tetrafluoroethylene) (Teflon)-lined autoclave in an oven after ultra-sounding for 5 min. Then the reaction was performed with a ramp rate of 8 °C/min at 80 °C for 7 h.

Synthesis of yellow CQDs (y-CQDs): 2,7-naphthalenediol (0.02 g) and $H_2SO_4$ (500 μL) were added in 7 mL of ethanol, the mixture solution was transferred into a poly(tetrafluoroethylene) (Teflon)-lined autoclave in an oven after ultra-sounding for 5 min. Then the reaction was performed with a ramp rate of 3 °C/min at 220 °C for 7 h.

Synthesis of orange CQDs (o-CQDs): 2,7-naphthalenediol (0.06 g) and acetic acid (260 μL) were added in 10 mL of ethanol, the mixture solution was transferred into a poly(tetrafluoroethylene) (Teflon)-lined autoclave in an oven after ultra-sounding for 5 min. Then the reaction was performed with a ramp rate of 6 °C/min at 200 °C for 12 h.

Synthesis of orange red CQDs (or-CQDs): 2,7-naphthalenediol (0.04 g) and urea (0.02 g) were dissolved in 10 mL of ethanol, the mixture solution was transferred into a poly(tetrafluoroethylene) (Teflon)-lined autoclave in an oven after ultra-sounding for 5 min. Then the reaction was performed with a ramp rate of 6 °C/min at 200 °C for 12 h.

Synthesis of red CQDs (r-CQDs): 2,7-naphthalenediol (0.05 g) and EDA (26 μL) were added in 10 mL of ethanol, the mixture solution was transferred into a poly(tetrafluoroethylene) (Teflon)-lined autoclave in an oven after ultra-sounding for 5 min. Then the reaction was performed with a ramp rate of 5 °C/min at 200 °C for 12 h. All the CQDs solution were filtered with a 0.22 μm microporous membrane under the reaction cooled down to room temperature. Subsequently, the CQDs were removed solvent with a rotary evaporator, and dialyzed against a dialysis bag (MW: 3500 Da) for one week. The CQDs powders were further vacuum drying at 80 °C after dialyzing.

### Offline testing of the PL wavelength and PLQY of ML-guided CQDs synthesis
The optical properties of CQDs, including their PL wavelength and PLQY, were meticulously characterized using the Horiba Duetta spectrometer. The PLQY of CQDs measurement referred to our previous work[21,51]. p-CQDs, b-CQDs, and g-CQDs choose quinine sulfate (standard PLQY of 0.58, conditions for PLQY measurement are 0.1 M $H_2SO_4$ aqueous solution, excitation = 350 nm) as the reference solution. c-CQDs and dc-CQDs choose rhodamine 101 (standard PLQY of 1, conditions for PLQY measurement are ethanol, excitation = 450 nm) as the reference solution. yg-CQDs and y-CQDs choose Rhodamine 6G (standard PLQY of 0.95, conditions for PLQY measurement are water, excitation = 488 nm) as the reference solution. o-CQDs choose rhodamine B (standard PLQY of 0.31, conditions for PLQY measurement are water, excitation = 514 nm) as the reference solution. or-CQDs choose cresyl violet (standard PLQY of 0.53, conditions for PLQY measurement are methanol, excitation = 580 nm) as the reference solution. r-CQDs choose Cy5 (standard PLQY of 0.27, conditions for PLQY measurement are PBS, excitation = 620 nm) as the reference solution. The absorbance (absorbance less than 0.1) of CQDs and different reference solutions at their respective optimal excitation wavelengths were tested, as well as the peak integration area of CQDs and different reference solutions in the range of 250–800 nm. PLQY is calculated using to the following formula:

$$PLQY = PLQY_r \times \frac{F}{F_r} \times \frac{A_r}{A} \times \left(\frac{\eta}{\eta_r}\right)^2 \quad (3)$$

In the formula, F-fluorescence emission peak area; A-absorbance at excitation wavelength; η-the refractive index of the solvent; r-reference substance.

### XGBoost
XGBoost, or Extreme Gradient Boosting, is a ML model that is capable to generalize on complex yet limited datasets[30,38,62]. It is a highly sophisticated and efficient implementation of the gradient boosting

algorithm, an ensemble technique that builds base estimators sequentially. Each subsequent base estimator aims to address the errors of its predecessors, and effectively integrating the weaker predictive base learners into a stronger ensembled model. Decision trees are predominantly employed as its base estimators. Specifically, our XGBoost models make prediction through an ensemble of M base estimators, i.e., decision trees $f_m$ with $i = 1, 2, \ldots, M$, and

$$\hat{y}_i = \sum_{m=1}^{M} f_m(x_i). \tag{4}$$

Given N training data $\{(x_i, y_i)\}_{i=1}^{N}$, XGBoost aims to minimize the objective function:

$$G(\theta) = \sum_{i=1}^{N} l(y_i, \hat{y}_i) + \sum_{m=1}^{M} \Omega(f_m), \tag{5}$$

where the first term is for reducing the discrepancy between predicted $(\hat{y}_i)$ and real values $(y_i)$, and the second term is to penalize the complexity of ensembled decision trees. During training, gradient boosting enforces the addition of a new tree at step t that minimize

$$g(\theta)^t = \sum_{i=1}^{N} l(y_i, \hat{y}_i^t) + \sum_{m=1}^{t} \Omega(f_m), \tag{6}$$

whereas

$$\hat{y}_i^t = \sum_{m=1}^{t} f_m(x_i) = \hat{y}_i^{t-1} + f_t(x_i). \tag{7}$$

XGBoost utilizes regularization technique to mitigate overfitting by penalizing model complexity.

## Structural characterization
Photographs of CQDs were photographed by a camera (D7000, Nikon). The optical properties of CQDs were characterized by Horiba Duetta. The time-resolved PL spectra were acquired using a Horiba FluoroMax. TEM images and HRTEM were performed by using a Japan Hitachi HT7700, and an USA FEI talos F200x G2, respectively. XRD data were achieved by Rigaku (SmartLab-II, Japan). Raman spectra were obtained by an Anton Paar Cora 5001 with 532 nm. FT-IR spectra were characterized by using a Thermo Scientific iS50. XPS spectra were assessed utilizing a Thermo ESCALAB 250Xi spectrometer. UPS spectra were obtained by ThermoFisher Nexsa.

## Data availability
Data that supports the findings of this study have been deposited in the GitHub repository and can be found with the following link: https://github.com/MS-w-ML/MOO-data.git. The data shown in Figs. 2–4 and Supplementary Figs. 3–24 are provided in the Source Data file. Source data are provided in this paper.

## Code availability
All machine learning codes used in this study have been deposited in the GitHub repository and can be found with the following link: https://github.com/MS-w-ML/MOO-code.git, or DOI at https://doi.org/10.5281/zenodo.11173329.

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

## Acknowledgements

This project was funded by the Shanghai Pujiang Program (Project No. 21PJD022 to L.W.), China Postdoctoral Science Foundation (Project No. 2023T160406 to H.G.), and the National Natural Science Foundation of China (Project No. 21901154 to L.W.). This project was also supported by the Ministry of Education, Singapore, under its Research Centre of Excellence award to the Institute for Functional Intelligent Materials (Project No. EDUNC-33-18-279-V12 to Z.L.), National Research Foundation, Singapore, under its AI Singapore Programme (AISG Award No: AISG2-GC-2023-009 to Z.L., B.T., and C.G.), as well as the Presidential Postdoctoral Fellowship of Nanyang Technological University (B.T.).

## Author contributions

H.G., Y.L., Z.L., and H.B. contributed equally to this work. L.W., Z.L., B.T., and C.G. conceived and supervised the project. H.G. and H.B. synthesized the samples and conducted structural and optical characterizations. Y.L., C.G., and B.T. designed the MOO framework, developed the algorithms and analyzed the results. Z.L., M.Z., and Z.W. assisted with data collection and analysis. H.G., Y.L., B.T., and L.W. co-wrote the manuscript. All authors discussed the results and commented on the final manuscript.

## Competing interests

The authors declare no competing interests.
