## [Peer Review File · Nature Communications]

Machine learning-guided realization of full-color high-quantum-yield carbon quantum dotsReviewer #1 (Remarks to the Author):

In this study, Guo et al. reported a novel multi-objective optimization strategy that integrates machine learning (ML) techniques with iterative experimental methodologies to guide the hydrothermal synthesis of carbon quantum dots (CQDs). The authors successfully demonstrated the application of ML to guide the CQDs synthesis, achieving both full-color fluorescence and high photoluminescence quantum yields through merely 63 experiments. The work is comprehensive, well-structured, and offers significant insights into the potential of ML in materials science. Thus, this manuscript is recommended to be published after addressing the following concerns:

1. Why was the XGBoost model chosen over other potential ML models?
2. The lifetimes of these full-color CQDs should be added.
3. Do these CQDs exhibit excitation-dependent behaviors?
4. It is essential to investigate whether the modification of the CQDs' surface with different functional groups has any influence on their luminescent properties.
5. Considering the importance of red and deep-red CQDs in various domains, a comparison with literatures on maximum PL wavelength would be valuable. Moreover, it would be intriguing to explore in the future work if ML could be applied to extend the PL wavelength into the near-infrared spectrum.

Reviewer #2 (Remarks to the Author):

This article reports the use of a machine learning technique (XGBoost) to learn and predict multiple properties from an experimental database. They target and succeed at designing a carbon quantum dot with full-color fluorescence and high PL quantum yield. This is remarkable because the database is very small including only 63 experiments. The approaches are well described, and the results appear to be correct.

My only concern is that the full database is not clearly presented in the SI, nor was it obvious how to find it on an openly available database. Therefore my only major recommendation is that the full database should be provided in a digital form (at minimum with comma-separated values (CSV)).

Reviewer #3 (Remarks to the Author):

This is an interesting paper that reports a guided method for developing efficient carbon dots that emit light across the visible spectrum. The carbon dots obtained by the authors' approach compare very favourably in terms of their photoluminescence quantum yield with carbon dots reported elsewhere in the literature, with emission wavelengths spanning the full visible spectrum.

The automated identification of reaction conditions needed to obtain chemical products with target properties is an active area of research, and the work described here is potentially a valuable contribution to the field. However, I have some concerns about the paper as it is currently written. In particular, in its current form, it is difficult to understand precisely what the authors have done. Much relevant information is given, but it is split across the main text, the methods section and the SI in such a way that it is difficult to understand the exact procedure the authors have used and hence make complete sense of the data obtained.

The current balance of the paper does not seem correct. The authors spend a relatively short amount of space describing the experimental and theoretical details of their approach, and devote a substantial amount of space in the main paper describing the characterization of properties that have not directly been optimized for. For most readers, the novelty and interest of the work is likely to lie in the automated procedures used to identify the optimized reaction conditions. The subsequent characterisation of the product (beyond the directly optimised properties of emission wavelength and PLQY) would be better moved to the supporting information, releasing space in the main text to better describe the experimental procedure.

A few specific comments:

- 1) Experimental conditions are given for the final optimized particles but – given the focus of the paper – it is also important to describe the general procedure that is being optimized. The complete synthesis procedure should be described in detail, with a substantially expanded description of how the listed “descriptors” (t, T, C, Vc, S, Rr and Mp) are applied experimentally. Also, it is not clear to me whether the procedure is carried out manually according to algorithm-directed instructions or whether it has been fully automated to create a self-optimising reactor / self-driving lab.
- 2) The “unified objective function” needs further explanation. The description on page 4 is insufficient, and the authors’ reasoning for defining the objective function in the specified manner should be better explained, e.g. how was the numeric value “10” decided upon. Have the authors evaluated other forms of objective function? If so, what led them to use the form specified in the paper?
- 3) The use of the word “color” in the main text is confusing. It would be better to discuss the optimization in terms of the peak emission wavelength, giving the permissible wavelength range for each colour. Currently the information concerning wavelength range is relegated to a column in table 3 of the supporting information.
- 4) The “XG Boost” procedure is not adequately described. More detail is needed.
- 5) The machining learning aspects of the work are unclear. As described, it would appear that the authors are using a responsive-surface based procedure, which might better be described as “optimization” rather than “machine learning”. A fuller discussion of the algorithmic procedure is needed to clarify this point.
- 6) How is PLQY measured as the optimization proceeds? Are the carbon dots evaluated inline or offline?
- 7) The results presented in Fig. 2 are the most important results for the work, and should be discussed in much more detail. The data in Fig. 2 should be split across several separate figures with a more thorough discussion of what each plot is showing.
- 8) In the discussion, the authors suggest that the search procedure is highly efficient, allowing a parameter space containing some “17 million possible combinations” to be reliably searched using just 63 measurements in total. It seems much more likely that the bounds of the descriptors have (not unreasonably) been pre-constrained to provide easy access to high-performing particles and that, within this parameter space, the optimization landscape is rather smooth, allowing high-performing particles to be readily identified. Further discussion is needed on this point.
- 9) The authors do not acknowledge or reference the wider literature on self-optimising reactors outside the field of carbon dots, including for instance highly relevant papers on the optimization of quantum dots by e.g. Krishnadasan, Jensen and Abolhasani.

Overall, this is interesting work, but the paper is rather difficult to follow in its current form and requires substantial restructuring and rewriting to more clearly convey how the authors have achieved the promising results presented.

Reviewer #4 (Remarks to the Author):

The manuscript by Guo et al. reports a method about multi-objective optimization of hydrothermal synthesis carbon dots via machine learning algorithm. Specifically, they unify the objective function for two target optical properties of carbon dots, including full-color PL wavelength and high PLQY. Full-color fluorescent carbon dots with remarkably high PLQY have been achieved in this work. However, I cannot recommend publication of this manuscript in its current form due to lack of

details. My questions to improve the manuscript are presented below:

- 1, In this manuscript, the authors selected eight descriptors to represent the hydrothermal synthesis carbon dots. Why they choose eight descriptors needs to be discussed in detail.
- 2, To establish the initial training dataset, authors collected 23 samples. Why only 23 samples were selected? The author augmented these samples, but how to enhance it and how many samples were used for training after augmented were not given in the manuscript.
- 3, The author employed gradient boosting decision tree-based model (XGBoost) to predict PL wavelength and the PLQY. Related work has been published earlier by the authors, how to reflect the novelty of this manuscript.
- 4, The authors declare has unveiled hidden relationships between synthesis parameters and various target properties. However, the correlation matrix in Figure 2d can only illustrate the degree of influence of parameters on properties, but the specific relationship cannot be explained through the correlation matrix.
5. The author has always emphasized that the proposed method learns from limited and sparse data, but still uses the standard XGBoost method for prediction without making corresponding improvements. Therefore, its rationality cannot be effectively explained.

Response to Reviewers' Comments

Response to Referee 1

In this study, Guo et al. reported a novel multi-objective optimization strategy that integrates machine learning (ML) techniques with iterative experimental methodologies to guide the hydrothermal synthesis of carbon quantum dots (CQDs). The authors successfully demonstrated the application of ML to guide the CQDs synthesis, achieving both full-color fluorescence and high photoluminescence quantum yields through merely 63 experiments. The work is comprehensive, well-structured, and offers significant insights into the potential of ML in materials science. Thus, this manuscript is recommended to be published after addressing the following concerns:

Response: We sincerely appreciate the referee for carefully reviewing our work and providing us invaluable suggestions to improve on it. In the revised manuscript, we have made our best efforts to address your concerns. Point-by-point responses are listed below.

Comment 1: Why was the XGBoost model chosen over other potential ML models?

Response: Thank you for your insightful question. We selected XGBoost for several compelling reasons, primarily due to its exceptional capability to extract complex patterns and generalize well on small datasets. This characteristic is particularly crucial in the context of ML-guided material synthesis, where datasets are inherently small due to high costs and extensive time requirements associated with experimental work. For example, our proposed ML-driven multi-objective optimization (MOO) strategy starts with a modest dataset comprising only 23 synthesis data points.

In our investigation, we considered four candidate models known for their efficacy in handling small datasets within the realm of materials science: XGBoost, multi-layer perceptron (MLP), support vector machine (SVM) and polynomial fitting (POLYFIT). To evaluate the

performance of these models, we employed nested cross validation on a dataset concerning hydrothermal-grown CQDs, derived from our previous study. It is revealed that XGBoost outperforms the other three candidate models, demonstrating its generalizability across related datasets, as shown in Supplementary Fig. 3. Therefore, XGBoost was selected as the surrogate ML model over other candidate models.

Supplementary Fig. 3 | Nested cross validation results of four candidate ML models. The surrogate ML model selection results of nested cross validation on the hydrothermal synthesis dataset extracted from our previous study¹. As shown above, XGBoost outperforms MLP, SVM and POLYFIT²⁻⁶ in terms of mean coefficient of determination (r^2), mean squared error (MSE), Pearson and normalized discounted cumulative gain (NDCG). Therefore, XGBoost is chosen as the surrogate ML model to predict the target properties for this work.

Comment 2: The lifetimes of these full-color CQDs should be added.

Response: Thank you for your constructive suggestion. To address your comment, we have conducted time-resolved PL spectroscopy measurements for the full-color CQDs using a Horiba FluoroMax instrument.

Our detailed analysis of the time-resolved PL spectra has uncovered a significant trend: the monoexponential lifetimes of the CQDs progressively decrease from 8.6 ns (p-CQDs) to 2.3 ns (r-CQDs), as illustrated in the newly added Supplementary Fig. 8. This trend indicates a reduction in the lifetimes of CQDs as their PL wavelength shifts towards the red spectrum. Changes to the manuscript and SI are provided below:

Page 8: *Furthermore, a comprehensive investigation of the time-resolved PL spectra revealed a notable trend. The monoexponential lifetimes of CQDs progressively decreased from 8.6 ns (p-CQDs) to 2.3 ns (r-CQDs) (Supplementary Fig. 8). This observation signified that the lifetimes of CQDs diminishes as their PL wavelength experiences a shift towards the red end of the spectrum⁴⁷.*

Supplementary Fig. 8 | Optical performance of full-color fluorescent CQDs. Time-resolved PL spectra of p-CQDs (a), b-CQDs (b), c-CQDs (c), dc-CQDs (d), g-CQDs (e), yg-CQDs (f), y-CQDs (g), o-CQDs (h), or-CQDs (i) and r-CQDs (j), respectively.

Comment 3: Do these CQDs exhibit excitation-dependent behaviors?

Response: Thank you for your valuable question. The dependency of excitation and emission in CQDs has been comprehensively studied by three-dimensional fluorescence spectroscopy. The related discussion in the manuscript is provided below:

Page 8: *Excellent excitation-independent behaviors of the CQDs have been further revealed by the three-dimensional fluorescence spectra (Supplementary Fig. 7).*

Supplementary Fig. 7 | Three-dimensional fluorescence properties of full-color fluorescent CQDs. Three-dimensional fluorescence spectra of p-CQDs (a), b-CQDs (b), c-CQDs (c), dc-CQDs (d), g-CQDs (e), yg-CQDs (f), y-CQDs (g), o-CQDs (h), or-CQDs (i) and r-CQDs (j), respectively.

Comment 4: It is essential to investigate whether the modification of the CQDs' surface with different functional groups has any influence on their luminescent properties.

Response: Thank you for your insightful comment. We have conducted a thorough investigation into the relationship between the surface functionalization of CQDs and their luminescent properties. Utilizing Fourier-transform infrared spectroscopy analysis (Supplementary Fig. 13), it was discovered that various CQDs (including p-/b-/c-/dc-/g-/yg-/or- and r-CQDs) are characterized by the presence of -NH₂, C=O, and C-O functional groups. Distinctly, y-CQDs, also exhibit -OH, C=O, C-O, and -SO₃ groups, whereas O-CQDs are marked by -OH, C=O, and C-O groups. These findings indicate a complex relationship between the specific functional groups on CQDs and their luminescent properties, suggesting that the influence of surface functionalization is not directly linear.

Moreover, through XPS-based analysis of the atomic content ratio of electron-donating to electron-withdrawing groups, a notable trend is revealed: the PL wavelength of CQDs shifts towards the red spectrum as the ratio of electron-donating groups decreases. This intriguing observation suggests a significant correlation between the optical properties of CQDs and their structural composition.

Figure R1 | Dependence of the PL wavelength and electron-donating/electron-with drawing groups of CQDs.

Table R1 | The electron-donating/electron-with drawing groups ratios of CQDs in XPS spectra and elemental analysis.

Sampl es	Element al	At. %	Electron-donating groups (at.%)				Electron-withdrawing groups (at.%)				
			Amin e N	Pyrroli c N	Graphit ic N	C-O	Pyridin ic N	Oxidize d N	C=O	N= O	S=O
r- CQDs	C	82.14									
	N	6.78	2.14	2.15	0.83	2.71	1.65	-	8.37	-	-
	O	11.08									
or- CQDs	C	81.04									
	N	4.45	1.17	1.30	0.93	0.63	1.05	-	3.82	-	-
	O	14.50									
o- CQDs	C	83.47									
	O	16.53	-	-	-	7.26	-	-	9.27	-	-
y- CQDs	C	88.75									
			-	-	-	2.62	-	-	5.0	-	3.17

School of Materials Science and Engineering

	O	10.80									
	S	0.45									
yg-CQDs	C	79.61									
	N	5.10	1.79	1.39	0.95	8.8 6	0.97	-	6.4 4	-	-
	O	15.30									
g-CQDs	C	75.82									
	N	6.30	2.07	1.87	0.32	2.4 7	1.58	0.46	9.4 2	6. 0	-
	O	17.89									
dc-CQDs	C	81.32									
	N	4.83	3.96	0.86	0.45	2.3 6	2.14	-	11.4 9	-	-
	O	13.85									
c-CQDs	C	82.37									
	N	2.38	0.89	0.80	0.20	9.5 0	0.41	0.09	5.76	-	-
	O	15.26									
b-CQDs	C	77.70									
	N	9.95	4.28	3.15	1.15	6.3 2	1.36	-	6.03	-	-
	O	12.35									
p-CQDs	C	81.88									
	N	3.54	0.70	1.47	1.08	6.2 2	0.29	-	8.35	-	-
	O	14.57									

Comment 5: Considering the importance of red and deep-red CQDs in various domains, a comparison with literatures on maximum PL wavelength would be valuable. Moreover, it would be intriguing to explore in the future work if ML could be applied to extend the PL wavelength into the near-infrared spectrum.

Response: Thanks for your valuable comment. We have acknowledged the significance of red and deep-red CQDs across various fields by incorporating a comparison of recent progress of PL wavelengths of red CQDs in the newly added Supplementary Table 4.

Moreover, we are excited about the potential of leveraging ML techniques to explore the extension of PL wavelength of CQDs into near-infrared spectrum. Building on the proven effectiveness of our current study, we believe that this innovative approach has the potential to lead to groundbreaking advancements. By pushing the boundaries of current capabilities, we aim to facilitate significant scientific progress in the development of CQDs with extended PL wavelengths in the near future. Change to the SI are shown below:

Supplementary Table 4 | Recent progress of PL wavelengths of red CQDs.

Samples	PL wavelength (nm)	Ref.
r-CQDs	645	This work
r-CQDs	620	7
r-CQDs	622	8
r-CQDs	625	9
r-CQDs	635	10
r-CQDs	625	11

r-CQDs	620	12
r-CQDs	621	13
r-CQDs	639	14
r-CQDs	635	15

References

1. Han, Y. et al. Machine-learning-driven synthesis of carbon dots with enhanced quantum yields. *ACS Nano* **14**, 14761-14768 (2020).
2. Lu, S. et al. Accelerated discovery of stable lead-free hybrid organic-inorganic perovskites via machine learning. *Nat. Commun.* **9**, 3405 (2018).
3. Xue, D. et al. Accelerated search for materials with targeted properties by adaptive design. *Nat. Commun.* **7**, 11241 (2016).
4. Rumelhart, D. et al. Learning representations by back-propagating errors. *Nature* **323**, 533-536 (1986).
5. Yuan, R. et al. Accelerated discovery of large electrostrains in BaTiO₃-based piezoelectrics using active learning. *Adv. Mater.* **30**, 1702884 (2018).
6. Sun, S. et al. Accelerated development of perovskite-inspired materials via high-throughput synthesis and machine-learning diagnosis. *Joule* **3**, 1437-1451 (2019).
7. Zhang, Q. et al. Photoluminescence mechanism of carbon dots: triggering high-color-purity red fluorescence emission through edge amino protonation. *Nat. Commun.* **12**, 6856 (2021).
8. Medina-Lopez, D. et al. Interplay of structure and photophysics of individualized rod-shaped graphene quantum dots with up to 132 sp² carbon atoms. *Nat. Commun.* **14**, 4728 (2023).
9. Shi, Y. et al. Red phosphorescent carbon quantum dot organic framework-based electroluminescent light-emitting diodes exceeding 5% external quantum efficiency. *J. Am. Chem. Soc.* **143**, 18941-18951 (2021).
10. Wang, B. et al. Rational design of multi-color-emissive carbon dots in a single reaction system by hydrothermal. *Adv. Sci.* **8**, 2001453 (2020).

11. Zheng, Y. et al. *Multicolor carbon dots prepared by single-factor control of graphitization and surface oxidation for high-quality white light-emitting diodes. Adv. Optical Mater.* **9**, 2100688 (2021).
12. Madonia, A. et al. *Dye-derived red-emitting carbon dots for lasing and solid-state lighting. ACS Nano* **17**, 21274-21286 (2023).
13. Ding, H. et al. *Large scale synthesis of full-color emissive carbon dots from a single carbon source by a solvent-free method. Nano Res.* **15**, 3548-3555 (2021).
14. Chen, J. et al. *Controlled synthesis of multicolor carbon dots assisted by machine learning. Adv. Funct. Mater.* **33**, 2210095 (2023).
15. Sun, S. et al. *Tumor microenvironment stimuli-responsive fluorescence imaging and synergistic cancer therapy by carbon-dot-Cu²⁺ nanoassemblies. Angew. Chem. Int. Ed.* **59**, 21041-21048 (2020).

Response to Referee 2

This article reports the use of a machine learning technique (XGBoost) to learn and predict multiple properties from an experimental database. They target and succeed at designing a carbon quantum dot with full-color fluorescence and high PL quantum yield. This is remarkable because the database is very small including only 63 experiments. The approaches are well described, and the results appear to be correct.

My only concern is that the full database is not clearly presented in the SI, nor was it obvious how to find it on an openly available database. Therefore, my only major recommendation is that the full database should be provided in a digital form (at minimum with comma-separated values (CSV)).

Response: Thank you for your constructive feedback and the positive remarks on our work. Regarding your concern about the database accessibility, we have uploaded the dataset including all 63 data points to GitHub, with plans to make it publicly accessible upon the publication of our paper. Change to the manuscript are shown below:

Page 14: ***Data availability***

Data that supports the findings of this study have been deposited in the GitHub repository and can be found with the following link: <https://github.com/MS-w-ML/MOO-data.git>.

Response to Referee 3

This is an interesting paper that reports a guided method for developing efficient carbon dots that emit light across the visible spectrum. The carbon dots obtained by the authors' approach compare very favorably in terms of their photoluminescence quantum yield with carbon dots reported elsewhere in the literature, with emission wavelengths spanning the full visible spectrum.

The automated identification of reaction conditions needed to obtain chemical products with target properties is an active area of research, and the work described here is potentially a valuable contribution to the field. However, I have some concerns about the paper as it is currently written. In particular, in its current form, it is difficult to understand precisely what the authors have done. Much relevant information is given, but it is split across the main text, the methods section and the SI in such a way that it is difficult to understand the exact procedure the authors have used and hence make complete sense of the data obtained.

The current balance of the paper does not seem correct. The authors spend a relatively short amount of space describing the experimental and theoretical details of their approach, and devote a substantial amount of space in the main paper describing the characterization of properties that have not directly been optimized for. For most readers, the novelty and interest of the work is likely to lie in the automated procedures used to identify the optimized reaction conditions. The subsequent characterization of the product (beyond the directly optimized properties of emission wavelength and PLQY) would be better moved to the supporting information, releasing space in the main text to better describe the experimental procedure.

Overall, this is interesting work, but the paper is rather difficult to follow in its current form and requires substantial restructuring and rewriting to more clearly convey how the authors have achieved the promising results presented.

Response: Thank you for your insightful comment and the recognition of the potential

contribution our work offers to the field of CQDs and synthesis condition optimization. We appreciate your insights regarding the structure and clarity of our manuscript and acknowledge the importance of addressing these concerns to better communicate our findings and methodologies. In response to your comments, we have undertaken a substantial restructuring of the paper to ensure a clearer presentation of our methods and results. Major changes are summarized below:

- (1) Expatiating the algorithmic procedure of the ML-driven unified multi-objective optimization strategy for further clarification
- (2) Elaborating the synthesis process of hydrothermal-grown CQDs and the measurement of target optical properties (i.e., PL wavelength and PLQY)
- (3) Explaining further the rationales for selecting eight synthesis descriptors and guidelines for defining their value ranges

With these comprehensive revisions, guided by your valuable suggestions, our manuscript should now be suitable for publication in *Nature Communications*. We look forward to your feedback and are hopeful for the positive consideration of our revised submission.

Comment 1: Experimental conditions are given for the final optimized particles but – given the focus of the paper - it is also important to describe the general procedure that is being optimized. (1) The complete synthesis procedure should be described in detail, with a substantially expanded description of how the listed “descriptors” (t, T, C, Vc, S, Rr and Mp) are applied experimentally. (2) Also, it is not clear to me whether the procedure is carried out manually according to algorithm-directed instructions or whether it has been fully automated to create a self-optimising reactor/self-driving lab.

Response: Thanks for your valuable comment. We appreciate the opportunity to clarify these aspects of our work further:

(1) In response to your request for a more detailed description of the synthesis procedure, we have thoroughly revised both the main text and the SI to provide a comprehensive overview. Specifically, the main text now includes a general outline of the CQDs synthesis process, highlighting the essential steps and key descriptors (t , T , C , V_C , S , V_S , R_r , and m_p) in the experimental setup. This overview is designed to give readers a clear understanding of the experimental approach and the significance of each parameter. Additionally, in the SI, we schematically illustrated how these descriptors are precisely controlled during the synthesis (Supplementary Fig. 3), accompanied by detailed synthesis recipes for each type of CQD (Supplementary Text). These enhancements aim to provide a clear, step-by-step guide to our methodology, ensuring that the process is transparent and replicable. Changes to the manuscript and SI are shown below:

Page 4: *Briefly, the 2,7-naphthalenediol molecule along with catalysts such as H_2SO_4 , HAc, ethylenediamine (EDA) and urea, were adopted in constructing the carbon skeleton of CQDs during the hydrothermal or solvothermal reaction process (Supplementary Fig. 2). Different reagents (including deionized water, ethanol, N,N-dimethylformamide (DMF), toluene, and formamide) were used to introduce different functional groups into the architectures of CQDs, combined with other synthesis parameters, resulting in tunable PL emission. To establish the initial training dataset, we collected 23 CQDs synthesized under different randomly selected parameters. Each data sample is labelled with experimentally verified PL wavelength and PLQY (see “Methods” section).*

Supplementary Fig. 2 | Schematic diagram of the parameter regulation process for synthesizing CQDs. There are eight primary control parameters in the synthesis process, including: (1) mass of precursor, (2) type of catalyst, (3) volume/mass of catalyst, (4) type of solution, (5) volume of solution, (6) reaction temperature, (7) ramp rate and (8) reaction time. These control parameters are involved in different steps of the synthesis process depicted. The detailed value ranges of the synthesis parameters are listed in Supplementary Table 1.

Supplementary Text: Preparation of full-color fluorescence CQDs

Synthesis of purple CQDs (p-CQDs): 2,7-naphthalenediol (0.04 g) and urea (0.02 g) were dissolved in 10 mL of toluene, the mixture solution was transferred into a poly(tetrafluoroethylene) (Teflon)-lined autoclave in an oven after ultra-sounding for 5 min. Then the reaction was performed with a ramp rate of 8 °C/min at 180 °C for 12 h.

Synthesis of blue CQDs (b-CQDs): 2,7-naphthalenediol (0.04 g) and EDA (170 μ L) were added in 10 mL of H₂O, the mixture solution was transferred into a poly(tetrafluoroethylene) (Teflon)-lined autoclave in an oven after ultra-sounding for 5 min. Then the reaction was performed with a ramp rate of 7 °C/min at 200 °C for 11 h.

Synthesis of cyan CQDs (c-CQDs): 2,7-naphthalenediol (0.08 g) and urea (0.1 g) were dissolved in 4 mL of H₂O, the mixture solution was transferred into a poly(tetrafluoroethylene)

(Teflon)-lined autoclave in an oven after ultra-sounding for 5 min. Then the reaction was performed with a ramp rate of 8 °C/min at 160 °C for 4 h.

Synthesis of darkcyan CQDs (dc-CQDs): 2,7-naphthalenediol (0.06 g) and urea (0.06 g) were dissolved in 10 mL of H₂O, the mixture solution was transferred into a poly(tetrafluoroethylene) (Teflon)-lined autoclave in an oven after ultra-sounding for 5 min. Then the reaction was performed with a ramp rate of 8 °C/min at 220 °C for 12 h.

Synthesis of green CQDs (g-CQDs): 2,7-naphthalenediol (0.02 g) and urea (0.06 g) were dissolved in 10 mL of ethanol, the mixture solution was transferred into a poly(tetrafluoroethylene) (Teflon)-lined autoclave in an oven after ultra-sounding for 5 min. Then the reaction was performed with a ramp rate of 8 °C/min at 160 °C for 12 h.

Synthesis of yellow green CQDs (yg-CQDs): 2,7-naphthalenediol (0.06 g) and EDA (940 µL) were added in 10 mL of DMF, the mixture solution was transferred into a poly(tetrafluoroethylene) (Teflon)-lined autoclave in an oven after ultra-sounding for 5 min. Then the reaction was performed with a ramp rate of 8 °C/min at 80 °C for 7 h.

Synthesis of yellow CQDs (y-CQDs): 2,7-naphthalenediol (0.02 g) and H₂SO₄ (500 µL) were added in 7 mL of ethanol, the mixture solution was transferred into a poly(tetrafluoroethylene) (Teflon)-lined autoclave in an oven after ultra-sounding for 5 min. Then the reaction was performed with a ramp rate of 3 °C/min at 220 °C for 7 h.

Synthesis of orange CQDs (o-CQDs): 2,7-naphthalenediol (0.06 g) and acetic acid (260 µL) were added in 10 mL of ethanol, the mixture solution was transferred into a poly(tetrafluoroethylene) (Teflon)-lined autoclave in an oven after ultra-sounding for 5 min. Then the reaction was performed with a ramp rate of 6 °C/min at 200 °C for 12 h.

Synthesis of orange red CQDs (or-CQDs): 2,7-naphthalenediol (0.04 g) and urea (0.02 g) were dissolved in 10 mL of ethanol, the mixture solution was transferred into a poly(tetrafluoroethylene) (Teflon)-lined autoclave in an oven after ultra-sounding for 5 min. Then the reaction was performed with a ramp rate of 6 °C/min at 200 °C for 12 h.

Synthesis of red CQDs (r-CQDs): 2,7-naphthalenediol (0.05 g) and EDA (26 μ L) were added in 10 mL of ethanol, the mixture solution was transferred into a poly(tetrafluoroethylene) (Teflon)-lined autoclave in an oven after ultra-sounding for 5 min. Then the reaction was performed with a ramp rate of 5 $^{\circ}$ C/min at 200 $^{\circ}$ C for 12 h. All the CQDs solution were filtered with a 0.22 μ m microporous membrane under the reaction cooled down to room temperature. Subsequently, the CQDs were removed solvent with a rotary evaporator, and dialyzed against a dialysis bag (MW: 3500 Da) for one week. The CQDs powders were further vacuum drying at 80 $^{\circ}$ C after dialyzing.

(2) Regarding the execution of the synthesis and characterization processes, these were carried out manually, following conditions recommended by our ML model. By detailing the synthesis procedure in both the manuscript and SI, we hope to eliminate any ambiguity, offering readers a detailed understanding of how we achieved the optimization of CQDs synthesis.

Comment 2: The “unified objective function” needs further explanation. (1) The description on page 4 is insufficient, and the authors’ reasoning for defining the objective function in the specified manner should be better explained, e.g. how was the numeric value “10” decided upon. (2) Have the authors evaluated other forms of objective function? If so, what led them to use the form specified in the paper?

Response: Thank you for highlighting the need for further clarification on the unified objective function. To address your suggestions:

(1) We have expanded our explanation on page 4 of the revised manuscript. Changes to the manuscript are shown below:

Page 4-5: *To account for the varying importance of multiple desired properties, an effective strategy is needed to quantitatively evaluate candidate synthesis conditions in a unified manner. A MOO strategy has thus been developed that prioritizes full-color PL wavelength over PLQY enhancement, by assigning an additional reward when maximum PLQY of a color*

surpassing the predefined threshold for the first time. Given N explored experimental conditions, $\{(x_i, y_i^c, y_i^y \mid i = (1, \dots, N))\}$, x_i indicates the i -th synthesis condition defined by 8 synthesis parameters, y_i^c and y_i^y indicate the corresponding color label and yield (i.e., PLQY) given x_i ; $y_i^c \in \{c_1, c_2, \dots, c_M\}$ for M possible colors, $y_i^y \in [0, 1]$. The unified objective function is formulated as the sum of maximum PLQY for each color label, i.e., $\sum_{c_j} Y_{c_j}^{\max}$, where $j \in \{1, 2, \dots, M\}$ and $Y_{c_j}^{\max}$ is 0 if $\nexists y_i^c = c_j$; otherwise $Y_{c_j}^{\max} = \max_i \left[\left(y_i^y + R \cdot \mathbb{1}(y_i^y \geq \alpha) \right) \cdot \mathbb{1}(y_i^c = c_j) \right]$. $\mathbb{1}(\cdot)$ is an indicator function that output 1 if true, otherwise outputs 0. The term $R \cdot \mathbb{1}(y_i^y \geq \alpha)$ enforces a higher priority of full-color synthesis, where PLQY for each color shall be at least α ($\alpha = 0.5$ in our case) to have an additional reward of R ($R = 10$ in our settings). R can be any real value larger than 1 (i.e., maximum possible improvement of PLQY for one synthesis condition), to ensure the higher priority of exploring synthesis conditions for colors in which yield has not achieved α . We set R to 10, such that the tens digit of unified objective function's value clearly indicates the number of colors with maximum PLQYs exceeding α , and the units digit reflects the sum of maximum PLQYs (without the additional reward) for all colors. As defined by the ranges of PL wavelength in Supplementary Table 2, seven primary colors considered in this work are purple (< 420 nm), blue (≥ 420 and < 460 nm), cyan (≥ 460 and < 490 nm), green (≥ 490 and < 520 nm), yellow (≥ 520 and < 550 nm), orange (≥ 550 and < 610 nm), and red (≥ 610 nm), i.e., $M = 7$. Notably, the proposed MOO formulation unifies the two goals of achieving full color and high PLQY into a single objective function, providing a systematical approach to tune synthesis parameters for the desired properties.

(2) This unified objective function is the only form evaluated due to its bespoke design for simultaneously optimizing the dual objectives of high PLQYs and full-color PL wavelength into a prioritized manner. Its effectiveness was confirmed through experimental verification, achieving the development of full-color, high-PLQY CQDs within just 20 iterations, underscoring its suitability and efficiency for our specific optimization problem.

Comment 3: The use of the word “color” in the main text is confusing. It would be better to discuss the optimization in terms of the peak emission wavelength, giving the permissible wavelength range for each color. Currently the information concerning wavelength range is relegated to a column in table 3 of the supporting information.

Response: Thank you for your constructive advice. To address your concerns:

(1) The “peak emission wavelength” is indeed used as the primary metric for optimization. This choice is driven by the need for accuracy and standardization in characterizing the photoluminescence properties of CQDs. The peak emission wavelength serves as a precise and quantifiable measure, facilitating the detailed analysis and comparison of CQDs properties.

(2) We primarily use the term “color” to emphasize our objective of achieving full-color fluorescence by incorporating a spectrum of fluorescent hues, while concurrently optimizing the PLQY of CQDs. This approach enables a more efficient and vibrant fluorescence outcome, thereby enhancing the overall effectiveness of the technique.

To clarify the connection between color and peak emission wavelength, we have supplemented our discussion with additional information in the revised manuscript. This includes an expanded explanation in the main text, alongside Supplementary Table 2, which illustrates the range of peak emission wavelengths associated with each color. These amendments are designed to provide a clearer understanding of how specific wavelengths correlate with the perceived color of fluorescence, ensuring that our approach and its outcomes are comprehensively communicated. Changes to the manuscript and SI are shown below:

Page 4-5: *As defined by the ranges of PL wavelength in Supplementary Table 2, seven primary colors considered in this work are purple (< 420 nm), blue (≥ 420 and < 460 nm), cyan (≥ 460 and < 490 nm), green (≥ 490 and < 520 nm), yellow (≥ 520 and < 550 nm), orange (≥ 550 and < 610 nm), and red (≥ 610 nm), i.e., $M = 7$.*

Supplementary Table 2 | Value ranges of colors defined by PL wavelength.

Target Properties		Value Range
	PLQY	[0, 1]
Color	Purple	PL < 420 nm
	Blue	420 nm ≤ PL < 460 nm
	Cyan	460 nm ≤ PL < 490 nm
	Green	490 nm ≤ PL < 520 nm
	Yellow	520 nm ≤ PL < 550 nm
	Orange	550 nm ≤ PL < 610 nm
	Red	610 nm ≤ PL

Comment 4: The “XG Boost” procedure is not adequately described. More detail is needed.

Response: Thank you for your constructive advice. In response, we have expanded the description of the XGBoost (Extreme Gradient Boosting) in the Methods section of our revised manuscript to provide a clearer understanding of its role and implementation within our multi-objective optimization strategy, particularly highlighting its effectiveness in handling small datasets. Changes to manuscript are highlighted below:

Page 13: *XGBoost, or Extreme Gradient Boosting, is a ML model that is capable to generalize on complex yet limited datasets^{30,38,63}. It is a highly sophisticated and efficient implementation of the gradient boosting algorithm, an ensemble technique that builds base estimators sequentially. Each subsequent base estimator aims to address the errors of its predecessors,*

and effectively integrating the weaker predictive base learners into a stronger ensemble model. Decision trees are predominantly employed as its base estimators. Specifically, our XGBoost models make prediction through an ensemble of M base estimators, i.e., decision trees f_m with $i = 1, \dots, M$, and $\hat{y}_i = \sum_{m=1}^M f_m(x_i)$. Given N training data $\{(x_i, y_i)\}_{i=1}^N$, XGBoost aims to minimize the objective function: $G(\theta) = \sum_{i=1}^N l(y_i, \hat{y}_i) + \sum_{m=1}^M \Omega(f_m)$, where the first term is for reducing the discrepancy between predicted (\hat{y}_i) and real values (y_i), and the second term is to penalize the complexity of ensemble decision trees. During training, gradient boosting enforces the addition of a new tree at step t that minimize $g(\theta)^t = \sum_{i=1}^N l(y_i, \hat{y}_i^t) + \sum_{m=1}^t \Omega(f_m)$, whereas $\hat{y}_i^t = \sum_{m=1}^t f_m(x_i) = \hat{y}_i^{t-1} + f_t(x_i)$. XGBoost utilizes regularization technique to mitigate overfitting by penalizing model complexity.

Comment 5: The machine learning aspects of the work are unclear. (1) As described, it would appear that the authors are using a response-surface based procedure, which might better be described as “optimization” rather than “machine learning”. (2) A fuller discussion of the algorithmic procedure is needed to clarify this point.

Response: Thank you for your constructive feedback. Our approach indeed leverages ML as a core component of our MOO strategy. Specifically, we employ ML models, i.e., XGBoost, to predict the target properties of CQDs under various synthesis conditions. While our MOO strategy focuses on optimizing experimental conditions to achieve desired outcomes, it aligns with the principles of response surface methodology (RSM) in seeking to model and optimize processes. However, our approach distinguishes itself from traditional RSM in several ways:

(1) Integration of ML Models: Unlike conventional response surface-based methods, which typically rely on simpler statistical or mathematical models for optimization, our strategy incorporates advanced ML models for the prediction and optimization of multiple target properties simultaneously. This allows for a more nuanced understanding and optimization of the synthesis conditions for CQDs.

(2) Optimization of Multiple Properties: Our MOO strategy is designed to optimize multiple

properties concurrently, which is a departure from the typical focus of RSM on optimizing a single response or a straightforward combination of responses. This multi-objective focus is particularly suited to the complex nature of CQD synthesis, where properties such as emission wavelength and photoluminescence quantum yield (PLQY) are both critical and interrelated.

(3) Future Directions: Acknowledging the potential overlap with RSM principles, we are indeed interested in exploring how response surface-based procedures could inform or enhance our framework in future work. This could involve integrating experiment planning aspects of RSM to further refine the optimization process and improve the efficiency of identifying optimal synthesis conditions.

In summary, while our method shares the optimization goal with response surface-based approaches, the integration of ML for predicting complex, multi-dimensional outcomes represent a significant advancement. During revision, we have elaborated on the algorithm procedures of our ML-powered MOO. Change to the SI are provided below:

Supplementary Fig. 1 | Flowchart of ML-guided synthesis of panchromatic CQDs. *Initially, a training set is collected with random selected synthesis conditions, and each condition is labelled by PL wavelength/PLQY from manual experimentation and characterization. Color labels (y_i^c) are given from discretizing predicted PL wavelength as shown in Supplementary Table 2. The size of initial training set (i.e., $N = 23$) is jointly determined by two requirements:*

1) the minimum number of data points (i.e., 20) required to run cross validation of ML surrogate models for hyperparameter searching, 2) the need to verify the viability of synthesizing full-color CQDs with this synthesis search space. Upon the 23rd randomly selected synthesis condition, synthesized CQDs exhibited versatile fluorescent colors with longspan emission wavelength data, achieving both requirements. After collecting the initial training set, two ML surrogate models with best hyperparameters are trained for predicting PL wavelength (or colors) and PLQY respectively. The two trained models are then employed to predict on all unexplored synthesis conditions. Then the corresponding utility values of these unexplored conditions are computed based on MOO objective function, through assuming that this one condition will be selected and augmented to the existing dataset of size N. Out of all unexplored conditions, the two with highest expected improvement of utility values are recommended for experimental verification and characterization. If the maximum PLQYs in all seven colors are larger or equal to 50%, the iterative ML-driven MOO loops end, otherwise the iteration proceeds and the two newly synthesized data points are augmented into the training set (i.e., $N = N + 2$) for the next round of model training and synthesis condition recommendation.

Comment 6: How is PLQY measured as the optimization proceeds? Are the carbon dots evaluated inline or offline?

Response: Thank you for your constructive questions. The PLQY of CQDs is determined through offline testing, following their synthesis and characterization. In the Methods section of the revised manuscript, we have provided a detailed description of the procedure used for the preparation and subsequent PLQY measurement of CQDs. Changes to the revised manuscript are shown below:

Page 12-13: ***Offline testing of the PL wavelength and PLQY of ML-guided CQDs synthesis***

The optical properties of CQDs, including their PL wavelength and PLQY, were meticulously characterized using the Horiba Duetta spectrometer. The PLQY of CQDs measurement

referred to our previous work^{23,53}. *p*-CQDs, *b*-CQDs, and *g*-CQDs choose quinine sulfate (standard PLQY of 0.58, conditions for PLQY measurement are 0.1 M H₂SO₄ aqueous solution, excitation = 350 nm) as the reference solution. *c*-CQDs and *dc*-CQDs choose rhodamine 101 (standard PLQY of 1, conditions for PLQY measurement are ethanol, excitation = 450 nm) as the reference solution. *yg*-CQDs and *y*-CQDs choose Rhodamine 6G (standard PLQY of 0.95, conditions for PLQY measurement are water, excitation = 488 nm) as the reference solution. *o*-CQDs choose rhodamine B (standard PLQY of 0.31, conditions for PLQY measurement are water, excitation = 514 nm) as the reference solution. *or*-CQDs choose cresyl violet (standard PLQY of 0.53, conditions for PLQY measurement are methanol, excitation = 580 nm) as the reference solution. *r*-CQDs choose Cy5 (standard PLQY of 0.27, conditions for PLQY measurement are PBS, excitation = 620 nm) as the reference solution. The absorbance (absorbance less than 0.1) of CQDs and different reference solutions at their respective optimal excitation wavelengths were tested, as well as the peak integration area of CQDs and different reference solutions in the range of 250-800 nm. PLQY is calculated using to the following formula:

$$PLQY = PLQY_r \times \frac{F}{F_r} \times \frac{A_r}{A} \times \left(\frac{\eta}{\eta_r}\right)^2$$

In the formula: *F*-fluorescence emission peak area; *A*-absorbance at excitation wavelength; η -the refractive index of the solvent; *r*-reference substance.

Comment 7: The results presented in Fig. 2 are the most important results for the work, and should be discussed in much more detail. The data in Fig. 2 should be split across several separate figures with a more thorough discussion of what each plot is showing.

Response: Thank you for your insightful feedback. Fig. 2 indeed presents important results regarding the evaluation of the proposed MOO strategy. However, given the importance of all figures in our manuscript, they are carefully curated to present a coherent narrative:

- Fig.1: Provides an overview of the proposed ML-driven MOO strategy, setting the stage

by introducing our interdisciplinary research framework.

- Fig. 2: Focuses on evaluating our approach through data-related metrics, offering a quantitative analysis of the strategy's effectiveness.
- Fig. 3: Delivers an evaluation from a materials science perspective, showcasing direct experimental evidence of the synthesized materials recommended by ML.
- Fig. 4: Offers insights into the successful application of ML guidance in the synthesis process.

Given the distinct yet complementary evaluations presented in Figs. 2 and 3, we decide not to divide Fig. 2 into multiple figures. This decision was made to maintain clarity and ensure a streamlined presentation of our findings. To enhance reader comprehension and provide a more detailed exploration of Fig. 2's significance, we have enriched its discussion sections in the manuscript. Additionally, we have updated the section title related to Fig. 3 and Fig 4 to highlight its connection to Fig. 2 more clearly, emphasizing the holistic evaluation of our proposed strategy. Changes to the manuscript are shown below:

Page 6-8: *To assess the effectiveness of our ML-driven MOO strategy in the hydrothermal synthesis of CQDs, we employed several metrics, which were specifically chosen to ascertain whether our proposed approach not only meets its dual objectives but also enhances prediction accuracy throughout the iterative process. The unified objective function described above measures how well the two desired objectives have been realized experimentally, and thus can be a quantitative indicator of the effectiveness of our proposed approach in instructing the CQD synthesis. The evaluation output of the unified objective function after a specific ML-guided synthesis loop is termed as objective utility value. The MOO strategy improves the objective utility value by a large margin of 39.27% to 75.44, denoting that maximum PLQY in all seven colors exceed the target of 0.5 (Fig. 2a). Specifically, at iteration 7 and 19, the number of color labels with maximum PLQY exceeding 50% increases by one, resulting in an additional reward of 10 each time. Even on the seemingly plateau, the two insets illustrate that the maximally achieved PLQY are continuously enhanced. For instance, during iterations 8 to 11, the maximum PLQY for cyan emission escalates from 59% to 94%, and the maximum PLQY*

for purple emission rises from 52% to 71%. Impressively, our MOO approach successfully fulfilled both objectives within only 20 iterations (i.e., 40 guided experiments).

Fig. 2b reveals that the MOO strategy systematically explores the synthesis conditions for each color, addressing those have not yet achieved the designed PLQY threshold, starting with yellow in the first 5 iterations and ending with green in the last 5 iterations. Notably, within each quintet of 5 iterations, a singular color demonstrates an enhancement in its maximum PLQY. Initially, the PLQY for yellow surges to 65%, which is then followed by a significant rise in purple's maximum PLQY from 44% to 71% during the next set of 5 iterations. This trend continues with cyan and green, where the maximum PLQY escalates to 94% and 83% respectively. Taking into account both the training set (i.e., the first 23 samples) and the augmented dataset, the peak PLQY for all colors exceeds 60%. Several colors approach 70% (including purple, blue, and red), and some are near 100% (including cyan, green, orange). This further underscores the effectiveness of our proposed ML technique. A more detailed visualization of the PL wavelength and PLQY along each iteration is provided in Supplementary Fig. 5.

*The MOO strategy ranks candidate synthesis conditions based on ML prediction; thus, it is vital to evaluate the ML model's performance. Mean squared error (MSE) is employed as the evaluation metric, commonly used for regression, which is computed based on the predicted PL wavelength and PLQY from the ML models versus the experimentally determined values⁴⁵. As shown in Fig. 2c, MSE of PLQY drastically decreases from 0.45 to *approximately 0.15 within just four iterations* — a notable error reduction of 64.5%. The MSE eventually stabilizes *around 0.1* as the iterative loops progress. Meanwhile, MSE of PL wavelength remains consistently low, always under 0.1. MSE of PL wavelength is computed after normalizing all values to the range of zero to one for fair comparison, thus MSE of 0.1 signifies a favorable deviation within 10% *between the ML-predicted values and the experimental verifications*. This indicates that the accuracies of our ML models for both PL wavelength and PLQY consistently improve, with predictions closely aligning with actual values after enhanced learning from augmented data. This not only demonstrates the efficacy of our MOO strategy in optimizing multiple desired properties but also in refining ML models.*

To unveil the correlation between synthesis parameters and target properties, we further calculated the covariance matrix. As illustrated in Fig. 2d, the eight synthesis parameters generally exhibit low correlation among each other, indicating that each parameter contributes unique and complementary information for the optimization of the CQDs synthesis conditions. In terms of the impact of these synthesis parameters on target properties, factors such as reaction time and temperature are found to influence both PL wavelength and PLQY. This underscores the importance for both experimentalists and data-driven methods to adjust them with higher precision. Besides reaction time and temperature, PL wavelength and PLQY are determined by distinct sets of synthesis parameters with varying relations. For instance, the type of solution affects PLQY with a negative correlation, while solution volume has a stronger positive correlation with PLQY. This reiterates that, given the high-dimensional search space, the complex interplay between synthesis parameters and multiple target properties can hardly be unfolded without capable ML-integrated methods.

To visualize how the MOO strategy has navigated in the expansive search space (~20 million) using only 63 data samples, we have compressed the initial training, explored, and unexplored space into two dimensions by projecting them into a new reduced embedding space using *t*-distributed stochastic neighbor embedding (*t*-SNE)⁴⁶. As shown in Fig. 2e, discerning distinct clustering patterns by color proves challenging, which emphasizes the intricate task of uncovering the relationship between synthesis conditions and target properties. This complexity further underscores the critical role of a ML-driven approach in deciphering the hidden intricacies within the data. The efficacy of ML models is premised on the quality of training data. Thus, selecting training data that span as large search space as possible is particularly advantageous to models' generalizability³⁷. As observed in Fig. 2e, our developed ML models benefit from the randomly and sparsely distributed training data, which in turn encourage the models to further generalize to previously unseen areas in the search space, and effectively guide the searching of optimal synthesis conditions within this intricate multi-objective optimization landscape.

Comment 8: In the discussion, the authors suggest that the search procedure is highly efficient, allowing a parameter space containing some “20 million possible combinations” to be reliably searched using just 63 measurements in total. It seems much more likely that the bounds of the descriptors have (not unreasonably) been pre-constrained to provide easy access to high-performing particles and that, within this parameter space, the optimization landscape is rather smooth, allowing high-performing particles to be readily identified. Further discussion is needed on this point.

Response: Thank you for your valuable comment. To clarify, the bounds of synthesis parameters are constraint by synthesis equipment and safety requirements instead of expert guidance. We have expanded upon this discussion in the manuscript to better articulate how the predefined parameter bounds, rooted in practical constraints, contributed to the successful optimization process. Changes to the manuscript are highlighted in red below:

Page 3-4: *To minimize human intervention, the bounds of synthesis parameters are determined primarily by the constraints of the synthesis methods and equipment used, instead of expert intuition. For instance, in employing hydrothermal/solvothermal method to prepare CQDs, as the reactor inner pot is made of polytetrafluoroethylene material, the usage temperature should be ≤ 220 °C. Moreover, the capacity of the reactor inner pot used in the experiment is 25 mL, with a general guidance of not exceeding 2/3 of this volume for reactions. Therefore, in this study, the main considerations of experimental design are to ensure experimental safety and accommodate the limitations of equipment. These practical considerations naturally led to a vast parameter space, estimated at 20 million possible combinations, as detailed in Supplementary Table 1.*

This extensive range encompasses both promising and less favorable regions, with the latter either yielding low PLQY or failing to produce CQDs, as shown in our provided dataset.

Despite this challenge, our ML-enhanced MOO strategy demonstrated remarkable efficiency in navigating this space. Within just 63 experiments, as depicted in Fig. 2e of our manuscript, we were able to identify optimal synthesis conditions for producing CQDs with the desired properties.

Supplementary Table 1 | Value ranges of input synthesis parameters. *The value ranges of synthesis parameters are determined on the configuration of synthesis devices in the laboratories. Compared to our previous study of CQDs¹, we expand the list of candidate solutions and catalysts to cater for the enhanced complexity of the targeted problem involving multiple desired properties. Type of catalyst and type of solution are ranked by polarity, whereas lower polarity leads to smaller indexes. The precursor used in our experiments is 2,7-naphthalenediol molecule.*

Parameter	Min	Max	Increment	# of Possible Points
Reaction temperature (°C)	80	220	20	8
Reaction time (hr)	1	12	1	12
Ramp rate (°C/min)	2	8	2	4
Mass of precursor (g)	0.02	0.2	0.02	10
Types of catalyst	EDA, H ₂ SO ₄ , HAc, urea			4
Volume/mass of catalyst (μL)/(mg)	0	1000	20	51
Types of solution	ethanol, deionized water, DMF, toluene, formamide			5
Volume (mL)	2	10	2	5
Total # of Possible Points				19,584,000

Comment 9: The authors do not acknowledge or reference the wider literature on self-optimising reactors outside the field of carbon dots, including for instance highly relevant papers on the optimization of quantum dots by e.g. Krishnadasan, Jensen and Abolhasani.

Response: Thank you for your valuable comment. We have included relevant literatures as Ref. 31-33 in the revised manuscript. Changes to the manuscript are shown below:

Page 2: *As a cutting-edge technology within the field of artificial intelligence, ML has proven to be a valuable tool in identifying complex relationships between material descriptors and desired properties, especially in high-dimensional and complex search spaces^{29,30-33}.*

References

31. Farhan, R. et al. Dietary bioavailability of cadmium presented to the gastropod *Peringia ulvae* as quantum dots and in ionic form. *Environ. Toxicol. Chem.* **32**, 2621-2629 (2013).
32. Andrew, F. et al. Machine-learning-guided discovery of electrochemical reactions. *J. Am. Chem. Soc.* **144**, 22599-22610 (2022).
33. Bateni, F. Sina S et al. Smart dope: a self-driving fluidic lab for accelerated development of doped perovskite quantum dots. *Adv. Energy Mater.* **14**, 2302303 (2024).

Response to Referee 4

The manuscript by Guo et al. reports a method about multi-objective optimization of hydrothermal synthesis carbon dots via machine learning algorithm. Specifically, they unify the objective function for two target optical properties of carbon dots, including full-color PL wavelength and high PLQY. Full-color fluorescent carbon dots with remarkably high PLQY have been achieved in this work. However, I cannot recommend publication of this manuscript in its current form due to lack of details. My questions to improve the manuscript are presented below:

Response: We sincerely appreciate the referee for carefully reviewing our work and providing us invaluable suggestions to improve on it. In the revised manuscript, we have made our best efforts to address your concerns. Point-by-point responses are listed below.

Comment 1: In this manuscript, the authors selected eight descriptors to represent the hydrothermal synthesis carbon dots. Why they choose eight descriptors needs to be discussed in detail.

Response: Thanks for your valuable comment. The set of eight descriptors are selected to represent the hydrothermal synthesis of CQDs due to two reasons:

(1) Extraction of Key Parameters from the Synthesis Process: The standard hydrothermal/solvothermal approach to fabricate CQDs typically involves using a polytetrafluoroethylene reactor inner pot, which operates at temperatures up to 220 °C and has capacity of 25 mL, with reactions usually not exceeding two-thirds of this volume for safety purposes. The synthesis parameters, including precursor mass, solution type, solution volume, reaction temperature, reaction time, and ramp rate are fundamental to the outcome of the synthesis. Moreover, the choice of catalyst, encompassing both the type and the amount used, is crucial due to its significant impact on the synthesis results. These considerations led to the

identification of eight synthesis parameters for this study. In the revised manuscript, we have elaborated on the synthesis process of CQDs as well as the descriptor selection. Changes to the manuscript and SI are shown below:

Page 3-4: *Using a representative and comprehensive synthesis descriptor set is of vital importance in achieving the optimization of synthesis conditions³⁶. We carefully selected eight descriptors to comprehensively represent the hydrothermal system, one of the most common methods to prepare CQDs. The descriptor list includes reaction temperature (T), reaction time (t), type of catalyst (C), volume/mass of catalyst (V_C), type of solution (S), volume of solution (V_S), ramp rate (R_r), and mass of precursor (m_p). To minimize human intervention, the bounds of synthesis parameters are determined primarily by the constraints of the synthesis methods and equipment used, instead of expert intuition. For instance, in employing hydrothermal/solvothermal method to prepare CQDs, as the reactor inner pot is made of polytetrafluoroethylene material, the usage temperature should be ≤ 220 °C. Moreover, the capacity of the reactor inner pot used in the experiment is 25 mL, with a general guidance of not exceeding 2/3 of this volume for reactions. Therefore, in this study, the main considerations of experimental design are to ensure experimental safety and accommodate the limitations of equipment. These practical considerations naturally led to a vast parameter space, estimated at 20 million possible combinations, as detailed in Supplementary Table 1. Briefly, the 2,7-naphthalenediol molecule along with catalysts such as H_2SO_4 , HAc , ethylenediamine (EDA) and urea, were adopted in constructing the carbon skeleton of CQDs during the hydrothermal or solvothermal reaction process (Supplementary Fig. 2). Different reagents (including deionized water, ethanol, N,N -dimethylformamide (DMF), toluene, and formamide) were used to introduce different functional groups into the architectures of CQDs, combined with other synthesis parameters, resulting in tunable PL emission. To establish the initial training dataset, we collected 23 CQDs synthesized under different randomly selected parameters. Each data sample is labelled with experimentally verified PL wavelength and PLQY (see “Methods” section).*

Supplementary Fig. 2 | Schematic diagram of the parameter regulation process for synthesizing CQDs. There are eight primary control parameters in the synthesis process, including: (1) mass of precursor, (2) type of catalyst, (3) volume/mass of catalyst, (4) type of solution, (5) volume of solution, (6) reaction temperature, (7) ramp rate and (8) reaction time. These control parameters are involved in different steps of the synthesis process depicted. The detailed value ranges of the synthesis parameters are listed in Supplementary Table 1.

(2) Selection of Independent Features: The independence of these descriptors was validated through a correlation analysis, the results of which are depicted in Fig. 2d. This analysis revealed that the selected parameters generally exhibit low correlations with one another, indicating that each provides unique and complementary information essential for the comprehensive understanding and optimization of the CQD synthesis process. We have also included this discussion in the revised manuscript as shown below:

Page 7: *To unveil the correlation between synthesis parameters and target properties, we further calculated the covariance matrix. As illustrated in Fig. 2d, the eight synthesis parameters generally exhibit low correlation among each other, indicating that each parameter contributes unique and complementary information for the optimization of the CQDs synthesis conditions. In terms of the impact of these synthesis parameters on target properties, factors such as reaction time and temperature are found to influence both PL*

wavelength and PLQY. This underscores the importance for both experimentalists and data-driven methods to adjust them with higher precision. Besides reaction time and temperature, PL wavelength and PLQY are determined by distinct sets of synthesis parameters with varying relations. For instance, the type of solution affects PLQY with a negative correlation, while solution volume has a stronger positive correlation with PLQY. This reiterates that, given the high-dimensional search space, the complex interplay between synthesis parameters and multiple target properties can hardly be unfolded without capable ML-integrated methods.

Fig. 2 | Evaluation of full-color CQDs synthesis guided by ML-based MOO strategy. *a*, MOO's unified objective utility versus design iterations. *b*, Color explored with new synthesized experimental conditions. It shows that when high PLQY has been achieved for “red”, “orange”, and “blue” in the initial dataset, the MOO strategy purposefully enhances PLQYs for “yellow”, “purple”, “cyan”, “green” respectively in subsequent synthesized conditions in group of five. *c*, MSE between the predicted and real target properties. *d*, Covariance matrix for correlation among the 8 synthesis parameters and 2 target properties, i.e., PLQY and PL wavelength (PL λ). *e*, Two-dimensional t-SNE plot for the whole search space, including unexplored (circular points), training (star-shaped points) and explored (square points) conditions, where the latter two sets are colored by real PL wavelengths.

Comment 2: (1) To establish the initial training dataset, authors collected 23 samples. Why only 23 samples were selected? (2) The author augmented these samples, but how to enhance it and how many samples were used for training after augmented were not given in the manuscript.

Response: Thanks for your valuable comment. We would like to address your concerns in two parts:

(1) To minimize human intervention in the ML-guided synthesis cycles as shown in Supplementary Fig. 1, the initial training data needs to meet two conditions with as few experiments: i) at least 20 data points to enable cross validation for ML model training and hyperparameter tuning; ii) prove the feasibility of synthesizing full-color high-yield CQDs in this hydrothermal system. Upon randomly selecting and experimentally testing 23 synthesis conditions from the entire search space, the resulting CQDs exhibited a broad spectrum of emission wavelengths. This outcome not only confirmed the system's viability for producing diverse CQDs but also satisfied the initial prerequisites for our study. Consequently, this process allowed us to establish a well-founded initial dataset, paving the way for the commencement of ML-guided synthesis iterations.

(2) The initial training data samples were not augmented, but the dataset was augmented as ML-guided synthesis iterations proceed. As shown in Supplementary Fig. 1, in each iteration, new ML models are trained based all the entirety of the available dataset (i.e., N data points), and recommends two best synthesis conditions out of all the unexplored. Subsequently, these recommended conditions are experimentally validated and incorporated into the dataset (or augmented to the dataset), thereby expanding it for the subsequent training of ML models in the next iteration (resulting in N+2 data points). This iterative process systematically enriches the dataset, ensuring continuous refinement and optimization of the synthesis parameters.

Supplementary Fig. 1 | Flowchart of ML-guided synthesis of panchromatic CQDs. *Initially, a training set is collected with random selected synthesis conditions, and each condition is labelled by PL wavelength/PLQY from manual experimentation and characterization. Color labels (y_i^c) are given from discretizing predicted PL wavelength as shown in Supplementary Table 2. The size of initial training set (i.e., $N = 23$) is jointly determined by two requirements: 1) the minimum number of data points (i.e., 20) required to run cross validation of ML surrogate models for hyperparameter searching, 2) the need to verify the viability of synthesizing full-color CQDs with this synthesis search space. Upon the 23rd randomly selected synthesis condition, synthesized CQDs exhibited versatile fluorescent colors with longspan emission wavelength data, achieving both requirements. After collecting the initial training set, two ML surrogate models with best hyperparameters are trained for predicting PL wavelength (or colors) and PLQY respectively. The two trained models are then employed to predict on all unexplored synthesis conditions. Then the corresponding utility values of these unexplored conditions are computed based on MOO objective function, through assuming that this one condition will be selected and augmented to the existing dataset of size N . Out of all unexplored conditions, the two with highest expected improvement of utility values are recommended for experimental verification and characterization. If the maximum PLQYs in all seven colors are larger or equal to 50%, the iterative ML-driven MOO loops end, otherwise the iteration*

proceeds and the two newly synthesized data points are augmented into the training set (i.e., $N = N + 2$) for the next round of model training and synthesis condition recommendation.

Comment 3: The author employed gradient boosting decision tree-based model (XGBoost) to predict PL wavelength and the PLQY. Related work has been published earlier by the authors, how to reflect the novelty of this manuscript.

Response: Thank you for your valuable comments. In response, we would like to provide a more systematic comparison between this work and our previous publication [Han, Y. et al. Machine-learning-driven synthesis of carbon dots with enhanced quantum yields. *ACS Nano* **14**, 14761-14768 (2020)], highlighting the novelty of the current study:

(1) Concurrent Optimization of Multiple Target Properties: This research marks a significant advancement by employing a ML-driven MOO strategy to simultaneously direct the synthesis of materials towards multiple objectives. Unlike the prior study, which concentrated on optimizing a single attribute, i.e., enhancing the PLQY within a specific color range, this work ambitiously targets dual objectives. It aims to span the entire visible spectrum from purple to red while also achieving high PLQY. By integrating a unified multi-objective function, we successfully extend the PL wavelength coverage from 410 to 645 nm, with PLQY exceeding 60% for all colors. This dual-objective approach facilitates a broader investigation into the optical capabilities of CQDs, opening avenues for diverse applications.

(2) Reduced Human Intervention: This significantly reduces human involvement by initiating with a compact, randomly selected initial training set and subsequently iterating under ML guidance. Previous methodologies typically commence with extensive datasets derived from expert insights. Consequently, this study enhances the utility of ML guidance by demonstrating its ability to efficiently navigate the synthesis process with minimal initial data.

(3) Efficient Utilization of Limited Data Demonstrating a novel ML-based MOO strategy, this work begins with an initial dataset of merely 23 data points and accomplishes the set targets

within just 20 iterations, totaling 63 experiments. This approach starkly contrasts with earlier efforts, including our own, where achieving a single target property (PLQY) necessitated a much larger experimental volume (over 500 sets), entailing considerable time and labor. The current study showcases the significant potential of ML-guided synthesis to expedite the development of materials in the early stages.

This innovative approach not only enhances the efficiency and efficacy of the ML-driven material research, but also provides a valuable reference for future studies in the field of nanomaterials.

Comment 4: The authors declare has unveiled hidden relationships between synthesis parameters and various target properties. However, the correlation matrix in Figure 2d can only illustrate the degree of influence of parameters on properties, but the specific relationship cannot be explained through the correlation matrix.

Response: Thank you for your insightful suggestion. We would like to address your comment in two points:

(1) **Objective of Fig. 2d:** The primary aim of presenting Fig. 2d is twofold: firstly, to showcase the extent to which various synthesis parameters influence the target properties, and secondly, to validate the efficacy of our feature selection process. This figure illustrates that the eight chosen synthesis parameters operate generally independently, each contributing unique and complementary insights into the synthesis process. This independence is crucial for ensuring that the ML model can effectively discern and leverage the distinct impact of each parameter on the desired properties of the CQDs. We have added this discussion in the revised manuscript as shown below:

Page 7: *To unveil the correlation between synthesis parameters and target properties, we further calculated the covariance matrix. As illustrated in Fig. 2d, the eight synthesis parameters generally exhibit low correlation among each other, indicating that each*

parameter contributes unique and complementary information for the optimization of the CQDs synthesis conditions. In terms of the impact of these synthesis parameters on target properties, factors such as reaction time and temperature are found to influence both PL wavelength and PLQY. This underscores the importance for both experimentalists and data-driven methods to adjust them with higher precision. Besides reaction time and temperature, PL wavelength and PLQY are determined by distinct sets of synthesis parameters with varying relations. For instance, the type of solution affects PLQY with a negative correlation, while solution volume has a stronger positive correlation with PLQY. This reiterates that, given the high-dimensional search space, the complex interplay between synthesis parameters and multiple target properties can hardly be unfolded without capable ML-integrated methods.

(2) Exploration of Specific Parameter-Property Relationships: For a more in-depth analysis of how specific synthesis parameters correlate with particular target properties, we employ boxplots to dissect these relationships further (Fig. 4d). A comprehensive discussion on this topic is provided on page 10, under the section titled “ML Insights for CQDs Synthesis.” Here, we discussed how individual parameters affect the synthesis outcomes, offering a clearer understanding of the underlying mechanisms that govern the production of CQDs with desired characteristics.

Page 10-11: ***ML insights into CQDs synthesis***

Following the effective utilization of ML in thoroughly exploring the entire search space, we proceeded to conduct a systematic examination of 63 samples using box plots, aiming to elucidate the complex interplay between various synthesis parameters and the resultant optical properties of CQDs. As depicted in Fig. 4d, the synthesis under conditions of high reaction temperature, prolonged reaction time, and low-polarity solvents, tends to result in CQDs with a larger PL wavelength. These findings are consistent with the general observations in the literature, which suggest that the parameters identified above can enhance precursor molecular fusion and nucleation growth, thereby yielding CQDs with increased particle size and high PL wavelength^{47,53-56}. Moreover, a comprehensive survey of existing literature implies that precursors and catalysts, typically including electron donation and acceptance, aid in producing long-wavelength CQDs^{57,58}. Interestingly, diverging from traditional findings, we

successfully synthesized long-wavelength red CQDs under ML guidance, with 2,7-naphthalenediol containing electron-donating groups as the precursor and EDA known for its electron-donating functionalities as the catalyst. This significant breakthrough questions existing assumptions and offers new insights into the design of long-wavelength CQDs.

Concerning PLQY, we found that catalysts with stronger electron-donating groups (e.g., EDA) led to enhanced PLQY in CQDs, consistent with earlier observations made by our research team¹⁶. Remarkably, we uncovered the significant impact of synthesis parameters on CQDs' PLQY. In the high PLQY regime, strong positive correlations were discovered between PLQY and reaction temperature, reaction time, and solvent polarity, previously unreported in the literature⁵⁹⁻⁶². This insight could be applied to similar systems for PLQY improvement.

Aside from the parameters discussed above, other factors such as ramp rate, the amount of precursor, and solvent volume also influence the properties of CQDs. Overall, the emission color and PLQY of CQDs are governed by complex, non-linear trends resulting from the interaction of numerous factors. It's noteworthy to mention that the traditional methods used to adjust CQDs' properties often result in a decrease in PLQY as the PL wavelength redshifts^{4,47,52,55}. However, utilizing AI-assisted synthesis, we have successfully increased the PLQY of the resulting full-color CQDs to over 60%. This significant achievement highlights the unique advantages offered by ML-guided CQDs synthesis and confirms the powerful potential of ML-based methods in effectively navigating the complex relationships among diverse synthesis parameters and multiple target properties within a high-dimensional search space.

Fig. 4 | Morphological characterizations and relationship analysis between synthesis parameters and optical properties of full-color fluorescent CQDs. **a**, The lateral size and color of full-color fluorescent CQDs (inset: dependence of the PL wavelength and the lateral size of full-color fluorescent CQDs). **b-c**, High-resolution TEM images and the fast Fourier transform patterns of p-, b-, c-, g-, y-, o- and r-CQDs, respectively. **d**, Boxplots of PL wavelength (left)/PLQY (right) and 7 synthesis parameters of CQDs. V_C is excluded here as its value range is dependent on C , whose relationships with other parameters are not directly interpretable. The labels at the bottom indicate the minimum value (inclusive) for the respective bins, whereas the bins on the left are the same as the discretization of colors in Supplementary Table 2, the bins on the right are uniform.

Comment 5: The author has always emphasized that the proposed method learns from limited and sparse data, but still uses the standard XGBoost method for prediction without making corresponding improvements. Therefore, its rationality cannot be effectively explained.

Response: Thank you for raising this concern. In addressing the use of XGBoost within our methodology, it's important to clarify the strategic choices behind its application and the broader context of our approach:

(1) Application of XGBoost: We chose XGBoost for its robustness and the incorporation of regularization techniques, which are crucial for mitigating overfitting, especially in scenarios involving limited datasets. This choice is supported by evidence from our previous research [Tang, B. et al. Machine learning-guided synthesis of advanced inorganic materials. *Mater. Today* **41**, 72-80 (2020); Han, Y. et al. Machine-learning-driven synthesis of carbon dots with enhanced quantum yields. *ACS Nano* **14**, 14761-14768 (2020)], where XGBoost demonstrated strong generalization capabilities on small datasets. The regularization feature of XGBoost plays a pivotal role in enhancing the model's performance by penalizing complex models, thus favoring simpler, more generalizable models that perform better on unseen data.

(2) Comprehensive MOO Strategy: It's essential to recognize that XGBoost serves as a

surrogate model within a larger MOO framework, which is meticulously designed to learn from sparse data. This framework comprises four integral components: database construction, MOO formulation, MOO recommendation, and experimental verification (illustrated in Fig. 1). The effectiveness of our approach is not solely reliant on the predictive power of XGBoost but also on the strategic integration of these components to form a cohesive learning loop.

(3) Optimization for Sparse Data: Beyond the choice of XGBoost, our methodology incorporates several strategies specifically tailored to enhance learning from limited and sparse datasets:

- The careful selection of an initial training set, strategically distributed across the entire search space, significantly benefits the model's ability to generalize, reducing the need for extensive training data.
- The introduction of a unified objective function for dual target properties effectively merges two complex search spaces into a single, more navigable space. This innovation simplifies the challenge of exploring the search space, facilitating more efficient identification of optimal synthesis conditions.

In summary, the rationality behind employing XGBoost, complemented by our comprehensive MOO strategy and specific optimizations for sparse data, collectively bolsters the effectiveness of our proposed method. This integrated approach enables us to effectively harness the potential of limited datasets, underscoring the efficacy of our ML-driven MOO methodology in navigating the challenges associated with sparse and small datasets.

Fig. 1 | Workflow of ML-guided synthesis of CQDs with superior optical properties. It consists of four key components: database construction, MOO formulation, MOO recommendation, and experimental verification.

Reviewer #1 (Remarks to the Author):

The authors have solved my concerns well. No further revision is required.

Reviewer #3 (Remarks to the Author):

The authors have made considerable efforts to address the issues I raised in my previous review, satisfactorily answering most points, and I am happy to recommend acceptance in its present form.

Reviewer #4 (Remarks to the Author):

I agree that the issues raised earlier have been appropriately addressed and I consent to publication.

Reviewer #4 (Remarks on code availability):

The author has provided thorough elaboration on the relevant details within the manuscript.